# Precipitation induced by explosive volcanism on Mars and its implications for unexpected equatorial ice

**Saira S. Hamid** [1] ✉, **Laura Kerber**[2] **& Amanda B. Clarke**[1,3]

Explosive volcanism occurred on Mars during its early history (Noachian–Hesperian; ~4.1–3.0 Ga). Because of Mars' cold atmospheric temperatures, water released from explosive eruptions may precipitate as ice or ice-ash aggregates. This process may have supplied ice to equatorial regions, which contain high excess hydrogen and potential buried ice deposits. We simulate explosive volcanic eruptions using the Laboratoire de Météorologie Dynamique Generic Planetary Climate Model and find that up to ~5 meters of ice is delivered to the surface in only one high-magnitude eruptive event. This ice can persist for long periods if preserved by widespread cooling from volcanic sulfuric acid or by burial under dust or pyroclasts. Here we show that over time, explosive eruptions may have served as a recurring mechanism for delivering ice to the equator, explaining elevated ice content at low latitudes independent of obliquity.

Mars has an ice-rich surface[1–7], but water ice is expected to be in equilibrium with the atmosphere only poleward of ±30° under the present-day orientation of the planet's obliquity (i.e., 25.19°)[1–3,5,8]. However, epithermal neutron data from the Mars Odyssey Neutron Spectrometer (MONS)[2,4,9], the Mars Odyssey High Energy Neutron Detector (HEND)[10], and the ExoMars Trace Gas Orbiter's Fine Resolution Epithermal Neutron Detector (FREND)[5] reveal excess hydrogen in the upper meter of the surface in equatorial regions (between ±30°). Several possibilities exist for the presence of excess hydrogen: (1) adsorbed water onto regolith particles[2,5], (2) water incorporated into the mineral's crystal structure (i.e., hydrated minerals[5]), (3) OH and $H_2O$ located in the structure of salt hydrates[11], (4) small amounts of water ice in the pores between regolith particles[5], (5) hydrous alteration in an aqueous environment[12], (6) sulfate hydration in the shallow subsurface[13], (7) OH that is part of the structure of clays and trapped water between clay layers[2] and/or (8) water interacting with cations located in the pores of zeolite mineral structure[2]. Using the pixon method—a Bayesian image reconstruction technique that groups pixels into larger units called "pixons"—Wilson et al., (2018) generated a higher-resolution map of the distribution of near-subsurface hydrogen on Mars based on epithermal neutron data

from MONS. This interpretation of the MONS data revealed notably high levels of excess hydrogen—measured as water-equivalent hydrogen (WEH)—near the equator, particularly around Meridiani Planum (up to ~14 wt% WEH) and the Medusae Fossae Formation (MFF) (up to ~50 wt% WEH) (Fig. 1). The excess hydrogen in these areas may suggest the existence of bulk ice. These findings imply there could be an "oasis" of bulk ice in the equatorial regions—an unexpected result that could have significant implications for future human exploration. However, the question remains: what is the origin of this ice?

High obliquities (i.e., >45°) may have allowed ice to migrate from the polar regions and stabilize in lower latitudes[14–16]. Lower obliquities (i.e., <45°), which characterized most of Mars' history[17,18], result in limited equatorial ice accumulation[14]. Even when ice does relocate to equatorial latitudes from the polar or low-elevation regions, it primarily accumulates around the Tharsis Montes, Terra Sabaea, Tyrrhena Terra regions, and only some portions of the MFF likely due to their higher elevations and topography relative to surrounding regions[14,15]. Therefore, remobilized ice from changes in obliquity alone may not fully explain the inferred presence of water in the equatorial region.

[1]School of Earth and Space Exploration, Arizona State University, Tempe, AZ, USA. [2]Jet Propulsion Laboratory, California Institute of Technology, Pasadena, CA, USA. [3]Istituto Nazionale di Geofisica e Vulcanologia, Sezione di Pisa, Pisa, Italy. ✉e-mail: sshamid1@asu.edu

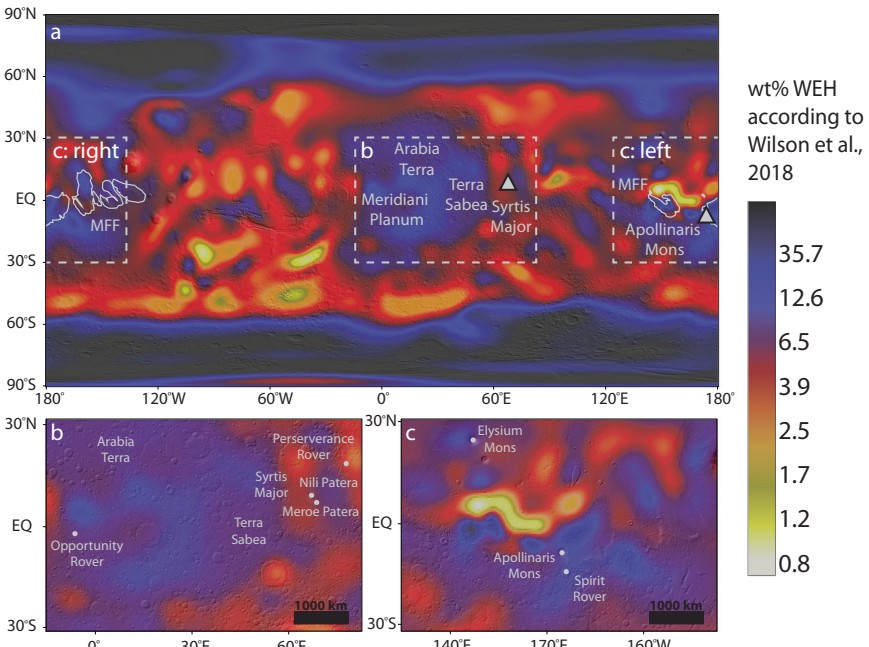

**Fig. 1 | Global distribution of near-surface hydrogen. a–c** Pixon reconstructed Mars Odyssey Neutron Spectrometer water equivalent hydrogen (WEH) data by Wilson et al., 2018 overlaid onto Mars Orbiter Laser Altimeter (MOLA) shaded relief map (illumination from the northeast)[141]. **a** Dotted boxes mark the locations of inset (**b** and **c**). **b** Excess hydrogen around Meridiani Planum and Syrtis Major. **c** Excess hydrogen around the Medusae Fossae Formation (MFF) and Apollinaris Mons.

Volcanism is a source of water for the atmosphere and can be emitted either passively or during an eruption[19,20]. Previous work modeling the release of water from passive volcanic degassing on Mars reveals that μm-thick deposits of frost accumulate throughout the equatorial region before quickly sublimating and migrating to existing cold traps (e.g., polar regions, Tharsis Montes, mid-latitudes)[21]. Explosive volcanic eruptions, however, can propel large pulses of water vapor from the volcano to higher levels of the atmosphere, which may cause it to interact with the atmosphere differently than passive degassing. The result may also be a deposit of an ash-ice mixture, or a layer of ice covered in ash, which may affect how long the ice remains[22].

Explosive volcanism occurs when magma is disrupted and ejected by expanding gases or external volatiles[23]. On Mars, the morphology of volcanic landforms suggest that explosive eruptions were more common in the planet's early geologic history before transitioning to more effusive eruptions, likely due to a wetter early mantle and/or interactions with external ice or liquid water present on the early Mars surface or near-surface[24–26]. These eruptions may have caused meteorological events that influenced the surface of Mars. For instance, terrestrial explosive volcanic eruptions are often likened to "dirty thunderstorms" due to their electrical activity, convective updrafts, and elevated water contents (from magmatic volatiles, interactions with external water, such as ground or surface water, and entrainment of moist tropospheric air)[27,28]. This water can encase volcanic ash, leading to phenomena like volcanic hail, graupel, snow, and rain[27]. Examples include the precipitation of volcanic hail in the 2009 eruption of Redoubt Volcano in Alaska[28], the development of volcanically induced thunderstorms and rain showers following the 1991 Mount Pinatubo, Philippines eruption[29], and the precipitation of volcanic snow from the 1980 Mount St. Helens eruption in Washington[30]. Since most ash particles are effective ice-forming nuclei[31], volcanic clouds and subsequent precipitation could have played a key role in shaping Mars' surface by delivering ash-ice mixtures or ice layers covered in ash. This ash can insulate and preserve underlying ice (discussed further in "Results": "The influence of sulfur and albedo on ice sublimation"), a phenomenon observed at Mount St. Helens, Mount Ruapehu in New

Zealand, and Villarrica Volcano in Chile[22]. Additionally, the accumulation of ash over snow or ice-covered soils can promote ground ice formation, as seen after the 2000 Hekla eruption in Iceland[22].

Explosive volcanic eruptions may have supplied ice to equatorial regions with high excess hydrogen (H), including to Meridiani Planum and the MFF. Meridiani Planum is a plain located at 0.04°S, 3.14°W near the hemispheric dichotomy boundary (i.e., the topographic division in elevation and crustal thickness between the Northern Lowlands and Southern Highlands of Mars). This region sparked interest after the nearby discovery of hematite deposits, which suggest chemical precipitation from aqueous fluids and the presence of stable liquid water at the surface or near surface[32]. The MFF is an extensive geologic deposit characterized by discontinuous, low density, fine-grained, and easily erodible surfaces spread across an area of ~$2 \times 10^6$ km² (20% of the continental United States) along the martian hemispheric dichotomy boundary[33–38], making it one of the largest fine-grained deposits on Mars[35]. The origin of the MFF is enigmatic, with several authors preferring a volcanic origin involving ash fall given its friable, easily erodible nature[33,35–37,39], tendency to drape over pre-existing topography[37,40], and a dielectric constant that aligns with that of a low-density material[41].

The closest volcanic source to Meridiani Planum is Syrtis Major, a Late Noachian–Late Hesperian-aged (~3.7–3.0 Ga) volcano containing two calderas, Meroe and Nili Patera, with possible resurfacing events into the Amazonian[42,43]. Syrtis Major contains evidence of highly evolved magmas in Nili Patera[44], possible ignimbrite deposits[45], and possible olivine ash deposits[25], suggesting Syrtis Major hosted high-explosivity eruptions. Additional calderas have been proposed in northwest Arabia Terra that could have contributed to volcanic ash in this area[26].

The MFF is also surrounded by possible volcanic sources, including Olympus Mons and the Tharsis Montes to the east, the Elysium volcanoes and Cerberus Fissures to the north, and Apollinaris Mons, an isolated volcano situated in the center of the deposit along the hemispheric dichotomy. Of particular interest for this study is Apollinaris Mons, a Mid Noachian–Early Hesperian volcano (~3.9–3.5 Ga)[42,43,46], which has been proposed to be the source for the

MFF[37] on the basis of Hesperian age determinations for the MFF (~3.7–3.0 Ga)[40], the location of the volcano in the center of the deposit, and evidence of pyroclastic activity due to its friable nature and sharply incised valleys[46,47]. Like the MFF, water is also thought to have played a role in the evolution of Apollinaris Mons based on morphological features indicative of an ice-rich substrate, such as the presence of craters with fluidized ejecta (i.e., ejecta formed by impact into subsurface volatiles[48–51]). Apollinaris Mons may also contain extant water ice on the basis of high excess hydrogen as measured by MONS, HEND, and FREND[2,4,5,10].

Is it possible that excess hydrogen present in equatorial regions was sourced from explosive volcanic eruptions from Apollinaris Mons and Syrtis Major? We test this specific hypothesis alongside broader questions about how explosive volcanism governs precipitation and ice distribution on Mars. While other workers[52] used volcanic ash deposition as a proxy for volcanic water enhancement in regolith, we explicitly model the release of water into the atmosphere as the result of an explosive eruption while tracking ice deposition. We also evaluate the impact of volcanic sulfuric acid ($H_2SO_4$) emissions, given their strong climatic influence on Earth[53–55] and the unique sulfur enrichment of regions such as Meridiani Planum and the MFF[56,57].

We select mass eruption rates (MERs) and corresponding plume heights of eruptions based on basic principles of buoyant convective rise conservation relationships established by Morton et al., (1956)[58], which have been applied to and calibrated on eruption plumes on Earth[59–64]. Additional eruption and environmental parameters are prescribed according to expected conditions on early Mars. We vary these parameters systematically (Table 1), holding others at baseline values (Table 2), to test sensitivity and determine the conditions under which explosive eruptions could generate and preserve equatorial ice deposits.

In this study, we demonstrate that explosive volcanism on early Mars could have served as a robust mechanism for delivering and maintaining low-latitude ice. Using the Laboratoire de Météorologie Dynamique (LMD) Generic Planetary Climate Model (PCM)[14,65], we simulate explosive eruptions from Apollinaris Mons and Syrtis Major, showing that they can generate meters-thick equatorial ice deposits consistent with hydrogen-rich regions such as Meridiani Planum and the MFF. By quantifying both water and sulfuric acid release, we establish volcanically induced precipitation as a recurring and independent pathway for sustaining equatorial ice, complementing and extending obliquity-driven explanations.

## Results and discussion
### Eruption characteristics influence rates of ice precipitation and accumulation

Over the course of an explosive volcanic eruption from Apollinaris Mons and Syrtis Major, the cold atmospheric temperatures cause water vapor to condense into ice, leading to widespread ice precipitation (Fig. 2a). The highest rates of precipitation occur near the eruptive vent, averaging 0.005 kg m$^{-2}$ s$^{-1}$ ($5.4 \times 10^{-6}$ m s$^{-1}$ or about 0.5 m sol$^{-1}$ assuming an ice density of 920 kg m$^{-3}$) under baseline conditions (Fig. 2a and Supplementary Table 1). For context, rates of snowfall across the continent of Antarctica, a region that is climatologically analogous to cold conditions likely present on early Mars, can reach ~0.02–0.5 m yr$^{-1}$ [66]. As ice precipitation persists, surface ice deposits accumulate, and these accumulation patterns can be compared to equatorial regions with high excess hydrogen (H) (Fig. 1 vs. Fig. 2b–d).

The modeled ice distribution broadly agrees with areas of excess H, with maximum deposition of 1340–1430 kg m$^{-2}$ (~1.5–1.6 m; Supplementary Table 1) predicted around Apollinaris Mons and Syrtis Major under baseline conditions. Under baseline conditions, eruptions from Syrtis Major result in ice fields in the highland regions around the volcano, the MFF, north pole, and Tharsis Rise. Eruptions from

**Table 1 | Model input parameters**

| Parameter | Values |
|---|---|
| *Eruption parameters* | |
| Volcanoes | Apollinaris Mons, Syrtis Major |
| Water mass eruption rate[122] | $10^6$, $10^7$, $10^8$, $10^9$ kg s$^{-1}$ |
| $H_2SO_4$ mass eruption rate[53,67] | $10^5$ kg s$^{-1}$ |
| Plume height | 35, 45, 65 km |
| Eruption duration[64] | 1, 3, 5 sols |
| *Environmental parameters* | |
| Obliquity[18] | 0°, 25.19°, 37.62°, 45°, 60° |
| Ice albedo[14,147,149] | 0.5, 0.645, 0.95 |
| Cloud condensation nuclei[14,147] | $10^4$, $10^5$, $10^6$ g$^{-1}$ |
| Topography | Pre- and post-Tharsis bulge |
| Season of eruption | Spring ($L_S = 0°$) |
| | Summer ($L_S = 90°$) |
| | Fall ($L_S = 180°$) |
| | Winter ($L_S = 270°$) |
| | Perihelion ($L_S = 251°$) |
| | Aphelion ($L_S = 71°$) |
| *Fixed parameters* | |
| Model resolution | 128 × 115 × 20 |
| $H_2SO_4$ radius[91] | 0.3 μm |
| Dust visible optical depth[112] | 0.2 |
| Stellar flux[111] | 1024.5 W m$^{-2}$ |
| Mean surface pressure[105] | 1 bar |
| Eccentricity[18] | 0.069 |
| Longitude of perihelion[151] | $L_S = 251°$ |

The model resolution is 128 × 115, which corresponds to a rectangular grid cell size ~166 × 92 km in longitude × latitude at the equator. The atmosphere contains 20 vertical levels ranging from the surface up to an altitude of 95 km. CCN values are for both liquid and ice water clouds. $L_S$ corresponds to the solar longitude. The water mass eruption rate is taken from the total mass eruption rate assuming an $H_2O$ content of 1 wt% for Martian magmas.

**Table 2 | Baseline simulation parameters**

| Parameters | Values |
|---|---|
| Water mass eruption rate | $10^9$ kg s$^{-1}$ |
| Plume height | 45 km |
| Eruption duration | 3 sols |
| Season | Spring ($L_S = 0°$) |
| Obliquity | 37.62° |
| Ice albedo | 0.645 |
| Cloud condensation nuclei | $10^5$ kg$^{-1}$ |
| Topography | Post-Tharsis bulge |

As each parameter is tested, all other parameters are held constant at the values specified above so that each effect may be tested separately. $L_S$ corresponds to the solar longitude.

Apollinaris Mons lead to dispersal in regions extending north and south from the volcano to the poles, with the thickest ice deposits forming around the MFF. The proximity of the MFF to Apollinaris Mons, combined with the tendency of eruptions from Syrtis Major to deposit ice in this region, helps explain the higher excess hydrogen observed around the MFF (up to ~50 wt% WEH)[4] and suggests a potential source for the buried ice deposits proposed by Watters et al. (2024) based on radar data. Following an eruption, ice fields continue to expand as residual atmospheric ice precipitates over the subsequent year (Fig. 2c, d). Furthermore, if Apollinaris Mons and Syrtis Major erupted before the Tharsis bulge fully formed —toward the later stages of its development— even greater ice deposition would occur in equatorial regions, since the Tharsis volcanoes would not act as high-elevation traps for surface ice (Fig. 2d).

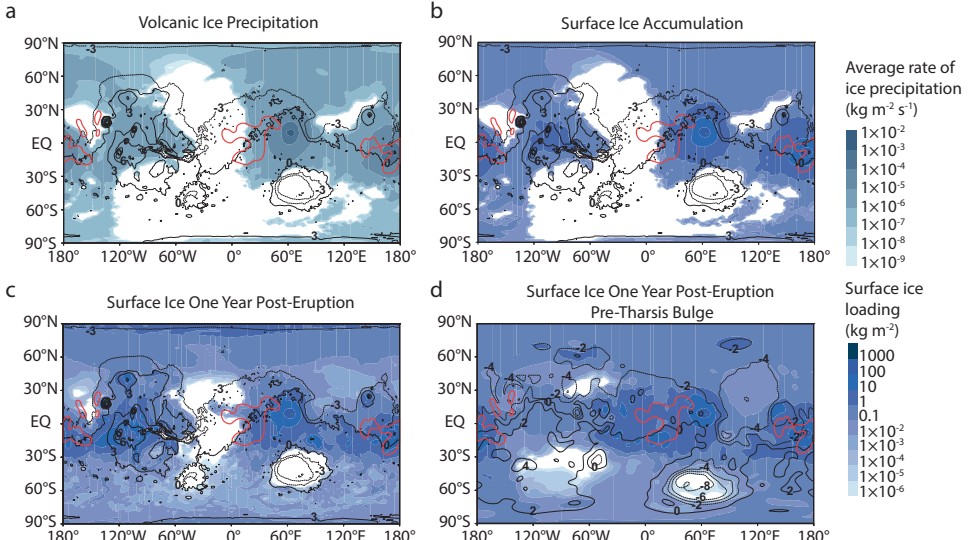

**Fig. 2 | Combined ice precipitation and surface ice distributions from Syrtis Major and Apollinaris Mons. a–d** Precipitation and surface ice distributions broadly overlap with equatorial regions containing high excess hydrogen (i.e., >10 wt% WEH; red lines). Eruptions from Syrtis Major (67.17°E, 9°N) and Apollinaris Mons (174.4°E, −8.5°S) are run according to baseline parameters listed in Table 2. **a**, **b** Ice precipitation and surface ice loading represent the resulting distributions at the end of the eruption. **c** Surface ice loading represents the resulting distribution one year post-eruption with present-day topography and **d** pre-Tharsis bulge topography. Modeled distributions are overlaid onto a cylindrical projection of Mars. Black lines represent MOLA (**a**–**c**) topographic contours and pre-Tharsis bulge (**d**) topographic contours, with numbers indicating contour elevations in kilometers.

The duration and MER are the primary controls on precipitation and surface ice accumulation (Figs. 3 and 4). Using Apollinaris Mons as a model case, we find that high water MERs (i.e., >$10^8$ kg s$^{-1}$; Fig. 3c, d) and prolonged eruption durations (i.e., 3–5 sols; Fig. 4b, c) produce the greatest agreement between modeled surface ice distributions and equatorial regions with high excess H, as they inject the most water into the atmosphere (e.g., a water MER of $10^9$ kg s$^{-1}$ sustained for 5 sols would result in a total released mass of $4.4 \times 10^{14}$ kg; Supplementary Table 2). Conversely, shorter durations and lower MERs inject less water into the atmosphere overall (e.g., a water MER of $10^6$ kg s$^{-1}$ sustained for 1 sol would result in a total released mass of $8.9 \times 10^{10}$ kg; Supplementary Table 2), resulting in more localized ice distributions centered around the eruption site.

Volcanic plume dispersal patterns are influenced by varying wind speeds present at different atmospheric levels. Consequently, the rate of ice precipitation and the loading and areal extent of subsequent surface ice accumulation are dependent upon the plume height. Given the behavior of plumes under the Martian atmosphere may behave differently compared to terrestrial plumes, we decouple the MER and plume height and test the sensitivity of ice distribution to each parameter separately (see "Methods": "Eruption parameters"). Using Apollinaris Mons as an example case, we find that plumes reaching higher altitudes are subjected to stronger zonal winds (winds that flow along lines of latitude) (Fig. 5), resulting in water being transported through the atmosphere for extended periods. These prolonged residence times allow ice fields to attain broader distributions compared to scenarios where the plume is injected at lower altitudes. However, lower plume heights cause ice to fall to the surface more readily, resulting in increased precipitation rates and up to ~4200 kg m$^{-2}$ or ~4.6 m of surface ice in the case of a 3-sol eruption at a water MER of $10^9$ kg s$^{-1}$ and a plume height of 35 km (Supplementary Table 1). The limits of the model are reached in particularly large eruptions with high plume heights (i.e., a water MER of $10^9$ kg s$^{-1}$ and a plume height of 65 km), as the rapid decrease in atmospheric temperatures caused by the high influx of water at such altitudes—and therefore low atmospheric pressures—exceeds the model's physical capabilities.

## The influence of obliquity, cloud condensation nuclei, and season on ice precipitation and accumulation

Volcanic precipitation can deliver ice to the equatorial regions marked by high hydrogen content across a wide range of obliquities, from 0° to 60° (Fig. 6). As obliquity increases, the poles receive more insolation than the equator, causing greater ice accumulation at lower latitudes and a stronger alignment between surface ice fields and areas with high excess hydrogen. However, regardless of obliquity, volcanogenic ice fields broadly align with the distribution of excess equatorial hydrogen, suggesting that surface hydrogen enrichment can be explained solely by volcanic sources.

Precipitation and surface ice accumulation are maximized if the abundance of ice-cloud nuclei, also referred to as cloud condensation nuclei (CCN, e.g., ash or dust) is moderate (i.e., the CCN = $10^5$ kg$^{-1}$; Supplementary Fig. 1). Local $H_2O$ cloud particle radii are based on the amount of condensed material, which is determined by the number of CCN per unit mass of air. Terminal fall velocities of these particles are then calculated from their radii and densities (see "Methods": "Water precipitation and deposition"). Using Apollinaris Mons as an example case, we find that changes in the number of CCN slightly changes precipitation rates throughout the course of the eruption and the resultant surface ice loading. If the CCN abundance is high (e.g., CCN = $10^6$ kg$^{-1}$), the ice is distributed among a larger number of nuclei leading to smaller cloud particle sizes overall. These smaller particle sizes result in slightly less ice precipitation on average and slightly thinner ice deposits around the volcano by the end of the eruption, whereas low and moderate amounts of CCN (i.e., $10^4$ and $10^5$ kg$^{-1}$) cause ice to be distributed among fewer nuclei, resulting in larger cloud particle sizes and slightly higher average precipitation rates and ice accumulation. For example, around Apollinaris Mons, precipitation rates are $4 \times 10^{-3}$ kg m$^{-2}$ s$^{-1}$ ($4.4 \times 10^{-6}$ m s$^{-1}$) for CCN = $10^6$ kg$^{-1}$ versus $5 \times 10^{-3}$ kg m$^{-2}$ s$^{-1}$ ($5.4 \times 10^{-6}$ m s$^{-1}$) for CCN = $10^4$ and $10^5$ kg$^{-1}$, with corresponding maximum accumulations of ~1200 kg m$^{-2}$ (~1.3 m) and ~1340 kg m$^{-2}$ (~1.5 m), respectively (Supplementary Table 1). Clouds that do not yet precipitate ice due to smaller particle sizes will linger in the atmosphere for longer periods until particle sizes increase and allow for subsequent precipitation or until the clouds dissipate.

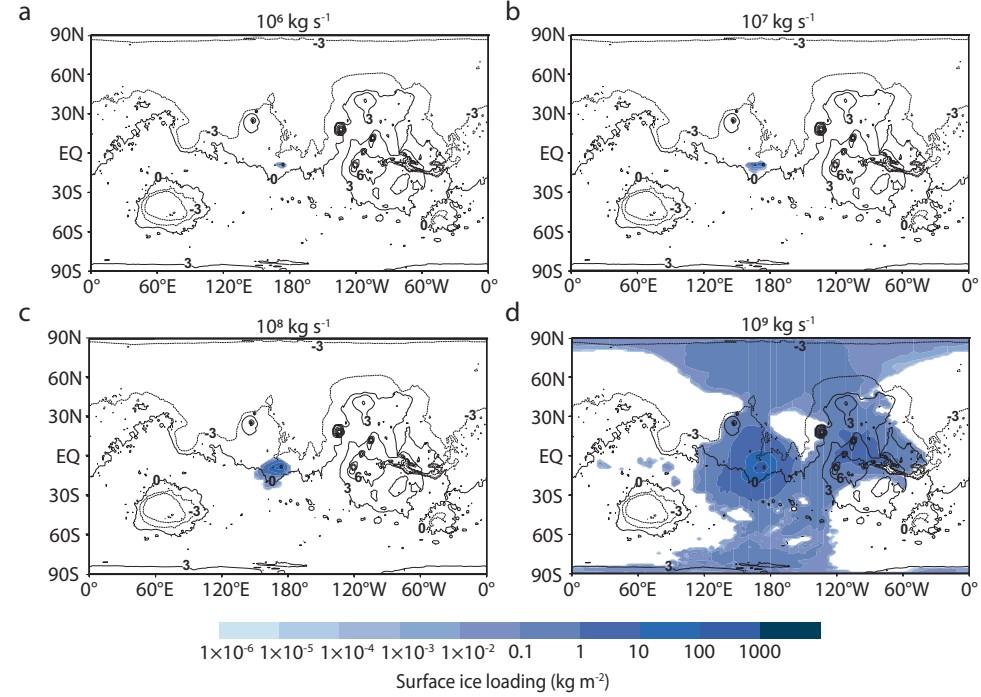

**Fig. 3 | The surface ice distribution is heavily influenced by the mass eruption rate. a–d** The eruptive vent of Apollinaris Mons is located at 174.4°E, −8.5°S. As the mass eruption rate is varied, other parameters are fixed at baseline conditions as listed in Table 2. Modeled distributions are overlaid onto a cylindrical projection of Mars. Black lines represent MOLA topographic contours, with numbers indicating contour elevations in kilometers. Ice fields represent resulting distributions at the end of the eruption.

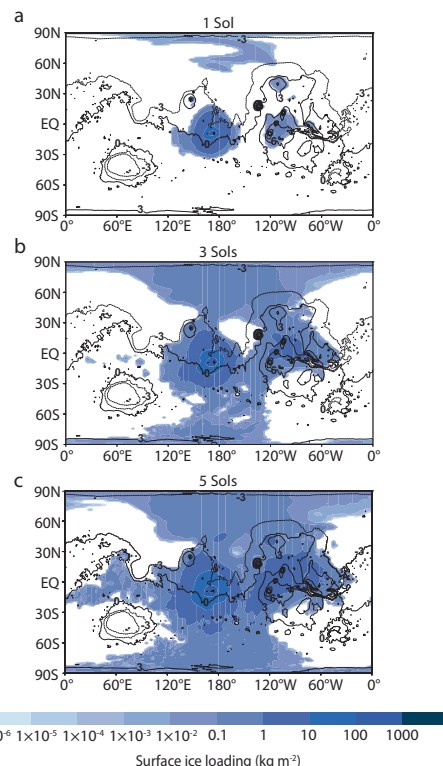

**Fig. 4 | The surface ice distribution is influenced by the eruption duration. a–c** The eruptive vent of Apollinaris Mons is located at 174.4°E, −8.5°S. As the duration is varied, other parameters are fixed at baseline conditions as listed in Table 2. Modeled distributions are overlaid onto a cylindrical projection of Mars. Black lines represent MOLA topographic contours, with numbers indicating contour elevations in kilometers. Ice fields represent resulting distributions at the end of the eruption.

The season of an eruption has minimal effect on surface ice distribution, as the ice deposited during eruptions surpasses sublimation rates caused by daily temperature fluctuations. Even a year after an eruption, ice distribution remains nearly identical across all seasons, including perihelion and aphelion (Supplementary Table 1).

### The influence of sulfur and albedo on ice sublimation

We simulate the release of sulfuric acid ($H_2SO_4$) into the atmosphere following an eruption at a rate analogous to the well-documented 1991 Mount Pinatubo eruption (i.e., $10^5 \, kg \, s^{-1}$)[53,67]. In the days following an eruption from Apollinaris Mons with plume height of 45 km, zonal winds transport $H_2SO_4$ through the atmosphere, leading to the dispersal of sulfuric acid around the planet's low-latitude regions (Fig. 7). Over the course of three months, $H_2SO_4$ gradually spreads to encircle the globe. $H_2SO_4$ from the eruption causes an immediate decrease in the amount of net radiation reaching the surface and plunges the planet into a global winter, resulting in lower surface temperatures throughout the year (except within Hellas Basin, in which averaged temperatures remain about the same) (Fig. 8). This climate change leads to enhanced protection of surface ice against sublimation even after the volcano has stopped erupting, lowering its sublimation rate to $1.3 \, kg \, m^{-2} \, yr^{-1}$ ($1.4 \times 10^{-3} \, m \, yr^{-1}$) compared to $14 \, kg \, m^{-2} \, yr^{-1}$ ($15 \times 10^{-3} \, m \, yr^{-1}$) in the same eruption but without $H_2SO_4$ present (Supplementary Fig. 2). Precipitation (e.g., ash-sulfur aggregation), chemical reactions, and atmospheric circulation will eventually filter the $H_2SO_4$ out of the atmosphere (e.g., after ~2 years in the case of the 1991 Mount Pinatubo eruption[53]). Assuming atmospheric circulation alone, we find that an eruption at a lower plume height (i.e., 35 km) causes $H_2SO_4$ to fall out of the atmosphere more readily, since it is injected closer to the surface than with taller plumes. When $H_2SO_4$ eventually does fall out of the atmosphere, the climate is expected to rebound to pre-eruptive climatic conditions and the remaining ice will be susceptible to normal sublimation rates.

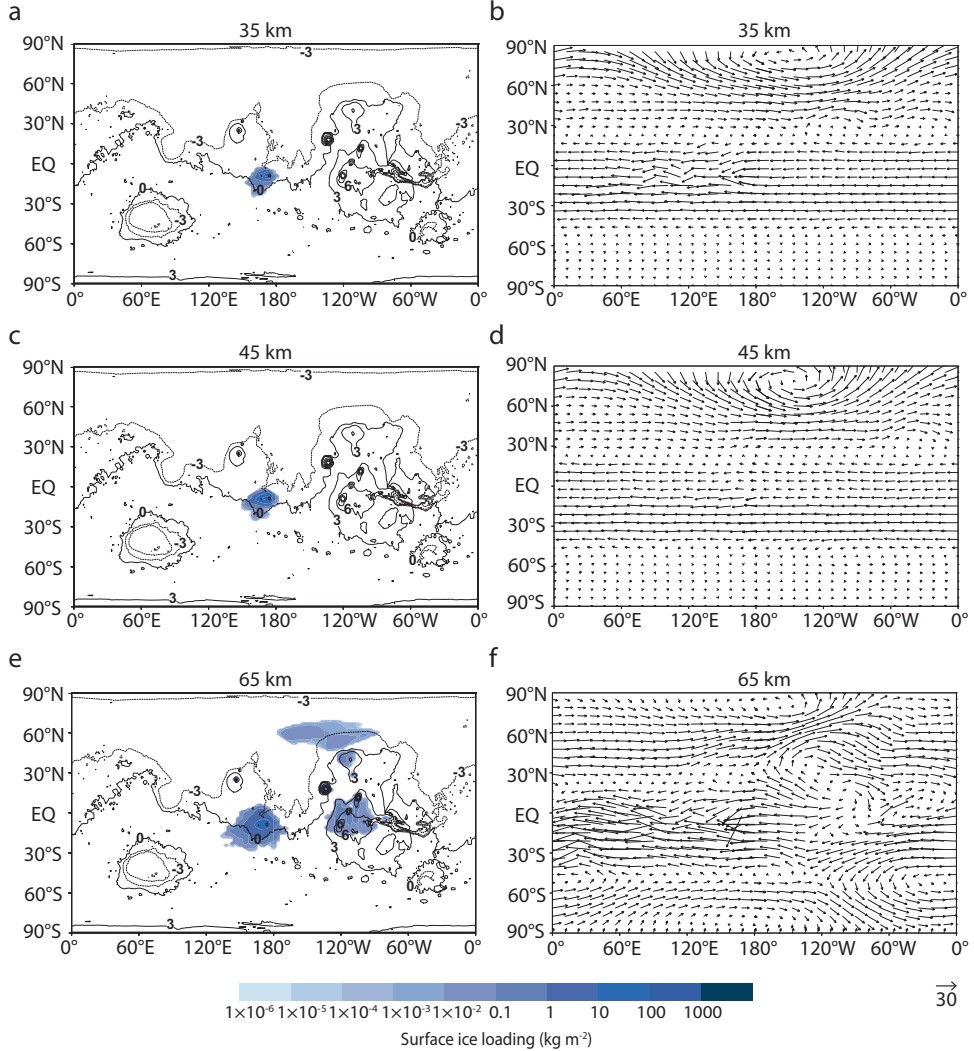

**Fig. 5 | Plume height and associated global wind patterns.** Variances in plume height (**a**, **c**, and **e**) and exposure to prevailing winds (in m s⁻¹) at the altitude of injection (**b**, **d**, and **f**) influence the loading and areal extent of surface ice deposits. Apollinaris Mons is located at 174.4°E, −8.5°S. The water mass eruption rate is $10^8$ kg s⁻¹ and other parameters are fixed at baseline values as listed in Table 2.

**a**, **c**, and **e** Modeled distributions are overlaid onto a cylindrical projection of Mars. Black lines represent MOLA topographic contours, with numbers indicating contour elevations in kilometers. Ice fields represent the resulting distribution at the end of the eruption. The reference arrow in the bottom right corner indicates the wind magnitude (in m s⁻¹) corresponding to the arrow length on the map.

Exposed surface ice is susceptible to sublimation from its exposure to solar radiation, especially after an explosive eruption stops supplying ice to the surface. Following the baseline eruption from Apollinaris Mons and Syrtis Major, surface ice continues to accumulate across the planet as lingering water in the atmosphere precipitates to the surface (Fig. 2c, d). The deposition of dust or ash onto layers of ice can influence its survival by changing the albedo and thus the amount of solar radiation absorbed by the ice[22]. However, we find that the ice albedo has minimal impact on the overall surface ice distribution, suggesting this ice could remain stable for long periods under a variety of albedo scenarios.

Sublimation rates may further decrease since ice that is co-deposited with ash may leave behind a lag deposit after the ice sublimates, which can preserve any remaining underlying ice by blocking access of water molecules to the atmosphere[1,22]. The resulting stratigraphy from repeated explosive eruptions throughout the lifetime of Apollinaris Mons and Syrtis Major may therefore contain ice-rich layers capped by ice-poor material. The thickness of these ice-rich layers ultimately depends on the duration, magnitude, and frequency of explosive events, which can vary widely depending on the specific volcano and its characteristics[68]. Many volcanoes produce multiple

scales of eruption at different temporal cycles. Moreover, as explosivity and magnitude increase, the recurrence frequency of volcanic eruptions tends to decrease[69].

## A volcanological case for surface ice enrichment

Volcanic precipitation provides a mechanism for ice to reach equatorial regions that exhibit high excess hydrogen. Consequently, it is important not to dismiss the possibility of surface ice accumulation through volcanic precipitation, as this process likely also played a crucial role in shaping ice-rich landforms around other volcanoes active on early Mars, such as Pityusa, Amphitrites, Malea, Tyrrhena, and Peneus Patera, and Hadriacus Mons[70–72].

The shift from explosive volcanic activity in early Mars to effusive activity later in Mars' history coincides with the decline in atmospheric pressure, a potential transition from a wetter to drier mantle, and the loss of surface water reservoirs[24]. However, low atmospheric pressure may still facilitate sporadic explosive eruptions. Reduced pressure allows for greater gas expansion and more efficient energy release, resulting in higher eruption velocities and more thorough magma fragmentation, leading to increased explosivity even with lower volatile concentrations[73]. Eruptions under a thinner atmosphere may have

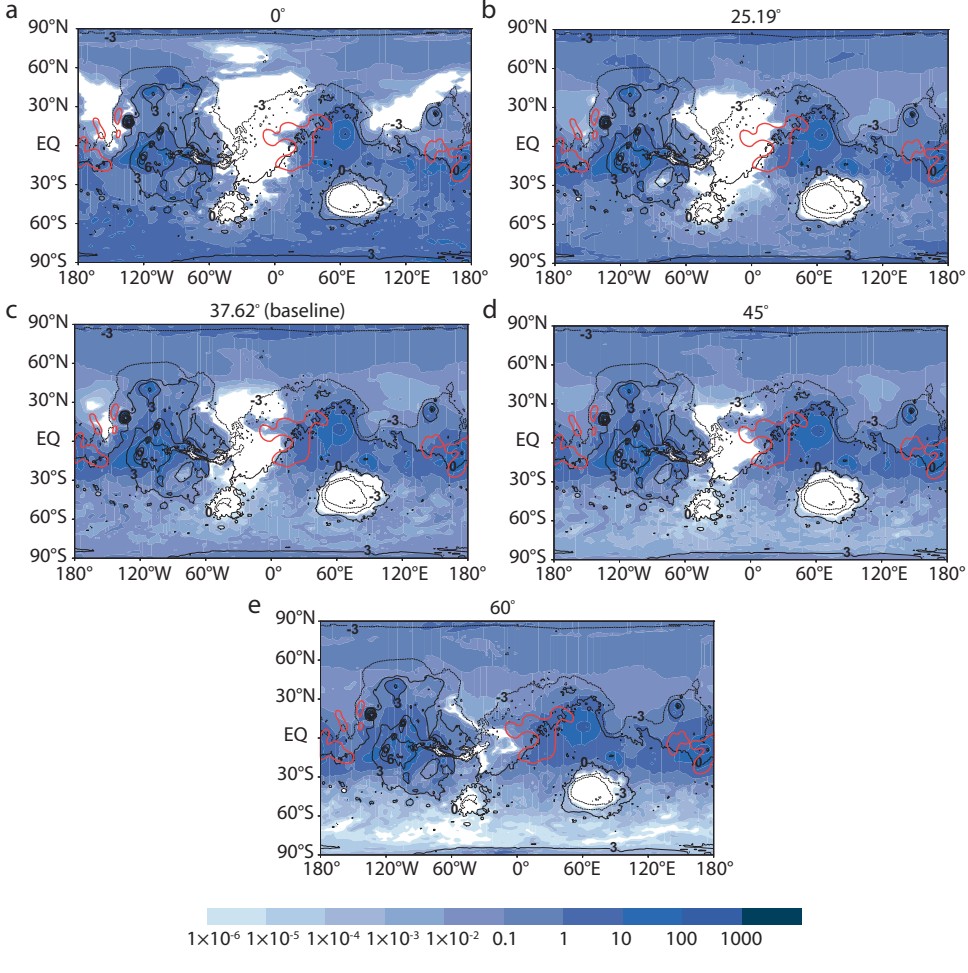

**Fig. 6 | Combined surface ice distributions across obliquity scenarios.**
**a**–**e** Modeled ice fields represent the resulting distribution one year following an eruption from Syrtis Major (67.17°E, 9°N) and Apollinaris Mons (174.4°E, −8.5°S). As the obliquity is varied, parameters are fixed at baseline conditions as listed in Table 2. Red lines represent equatorial regions containing high excess hydrogen (i.e., >10 wt% WEH). Modeled distributions are overlaid onto a cylindrical projection of Mars. Black lines represent MOLA topographic contours, with numbers indicating contour elevations in kilometers.

promoted more surface ice deposition due to the reduced capacity of the atmosphere to retain moisture. Additionally, the faster winds associated with the thinner atmosphere could have enabled broader dispersal of volcanic water[21]. We thus propose that younger shield volcanoes on Mars may have undergone episodic explosive activity, potentially contributing to ice-rich substrates and morphologies on both local and global scales. Potential examples include Elysium Mons[74,75], Hecates Tholus[76,77], Olympus Mons[78], Alba Mons[79], Arsia Mons[80], and the low shields within the Tharsis volcanic province[81].

Eruptions occurring throughout Mars' history, including under reduced atmospheric pressure, may have contributed to past aqueous activity. This includes the valley network period (and possibly paleo-lakes) formed around the Noachian-Hesperian boundary (-3.8–3.6 Ga[82]), minor valley formation continuing through the Hesperian and into the Early Amazonian (-2.8 Ga[82]), and hydrologic events linked to the formation of alluvial fans from the Hesperian to the Amazonian (-3.7–2.5 Ga[83]). In essence, explosive volcanic eruptions may have dramatically transformed Mars' surface over time, redistributing volatile resources without requiring changes in planetary obliquity. These insights about near-surface ice associated with volcanism and volcanic regions have significant implications for in-situ resource utilization during future human exploration, potential sites for sample return missions[84], and astrobiology, as ice deposits and volcanic areas are considered promising habitats for microbial life on Mars[85–88].

## Methods

### Explosive eruption simulations with a planetary climate model

The LMD Generic PCM, formerly known as the Generic Global Climate Model, can simulate a wide variety of planetary climate scenarios, including that of early Mars[14,89–91]. The PCM uses the LMDZ dynamical core, which is based on a finite-difference formulation of the primitive equations of geophysical fluid dynamics[65]. The PCM calculates the temporal evolution of variables that control the planetary climate at different points on a 3-dimensional grid spanning the Martian atmosphere. The PCM also includes a generalized correlated-k method that produces a database of coefficients to be used by radiative transfer calculations, which takes into account the absorption and scattering by the atmosphere, the clouds, and the surface[14]. Simulations are computed at a grid resolution of 128 × 115 cells, which corresponds to a rectangular grid cell size -166 × 92 km in longitude × latitude at the equator. In the vertical direction, the model is composed of 20 distinct atmospheric layers that spans from the surface up to an altitude of 95 km, assuming a pressure scale height of 8 km. The PCM uses hybrid sigma-pressure coordinates[65], in which sigma coordinates (ratio between pressure and surface pressure) are used near the surface and

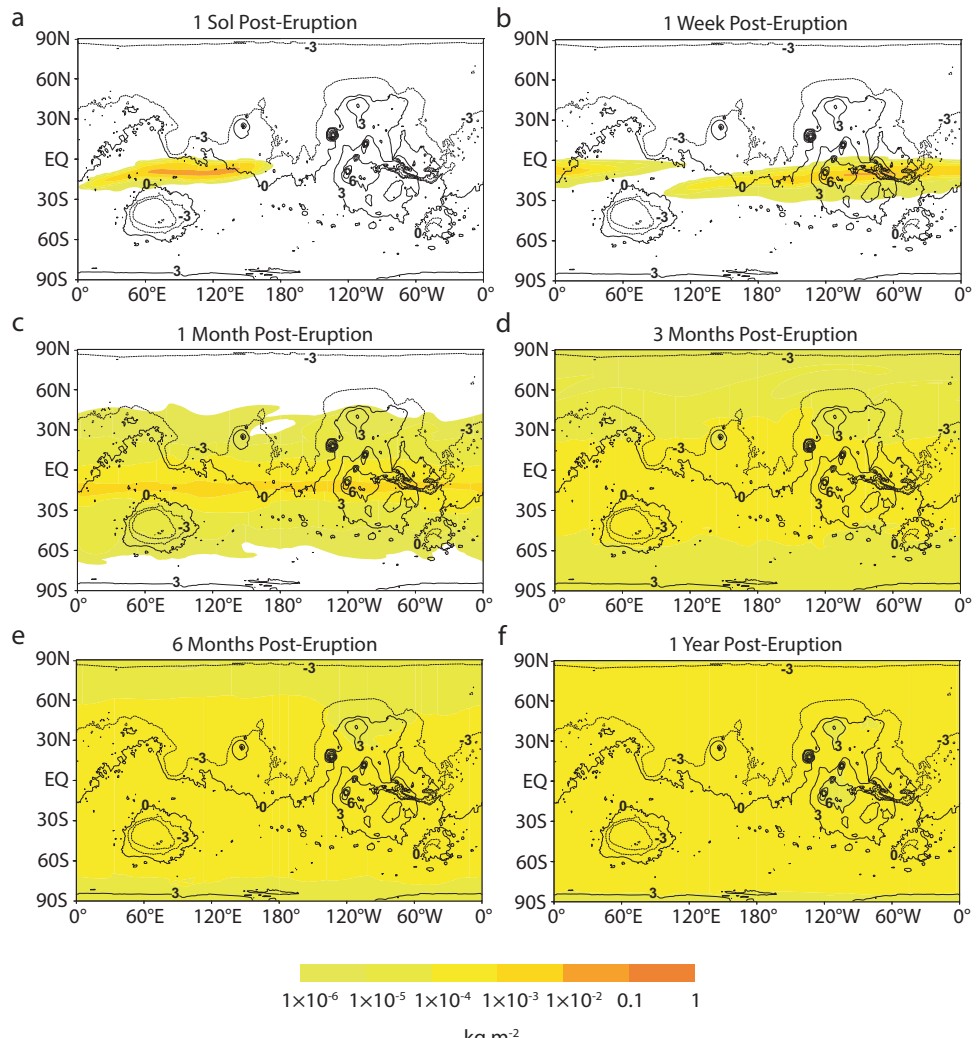

**Fig. 7 | Sulfuric acid disperses across the planet in the year following the eruption. a–f** Sulfuric acid ($H_2SO_4$) dispersal refers to the amount of $H_2SO_4$ in a vertical column over one grid cell from the surface to the top of the model (65 km) following an eruption from Apollinaris Mons (174.4°E. −8.5°S). Simulation is run for 1 year following the eruption under baseline conditions as listed in Table 2, except with a water mass eruption rate of $10^7$ kg s$^{-1}$ and an $H_2SO_4$ mass eruption rate of $10^5$ kg s$^{-1}$. Modeled distributions are overlaid onto a cylindrical projection of Mars. Black lines represent MOLA topographic contours, with numbers indicating contour elevations in kilometers.

are gradually shifted to purely pressure coordinates with increasing altitude. The distribution of the vertical layers is irregular, with more layers present at the ground level to enable greater precision. The first layer describes the first few meters above the ground, whereas the upper layers span several kilometers. This resolution is much greater than resolutions used in previous Generic PCM simulations[14,21,89], allowing for a more detailed representation of physical processes that govern the transport of volcanic water and sulfuric acid through the atmosphere. The initial state of simulations and input parameters for sensitivity tests are further described in Sensitivity Studies.

The LMD Generic PCM allows us to simulate the release of volcanic water and sulfuric acid from explosive convective plumes while also taking into account the localized meteorology that might influence dispersal and changes in wind speed and direction that take place as winds encounter downstream topographic features. Upon eruption from the vent, a volcanic plume is initially denser than the surrounding atmosphere. As the plume entrains and heats the ambient atmosphere, it becomes buoyant in the atmosphere and begins to rise convectively to the height of neutral buoyancy, where the majority of small ash particles are released[61,92]. We do not explicitly model convective

plumes or volcanic ash within the PCM. Instead, the effects of volcanic ash are represented indirectly through cloud condensation nuclei (CCN), (i.e., water-attracting particles) and ice albedo parameters, as atmospheric impurities like ash can promote cloud formation and alter the albedo of surface ice (discussed further in "Environmental parameters"). We follow the simplified approach outlined in Hamid et al., (2024, Eqs. 2 and 3) and release volcanic water and sulfuric acid at a specified MER, duration, and height (discussed further in "Eruption parameters"). We first find the mass of air in a single grid box in the PCM. Assuming hydrostatic equilibrium, the PCM calculates the mass of air in a grid box:

$$M_{air} = \frac{dPA}{g} \tag{1}$$

Where $dP$ is the change in pressure between the top and bottom of the grid box, $A$ is the horizontal area of the grid cell, and $g$ is the gravitational acceleration of Mars. To simulate volcanic eruptions, a fixed MER of volcanic water and sulfuric acid, $F_{MER}$ is then released into the grid point nearest the centroid coordinates of the Apollinaris Mons

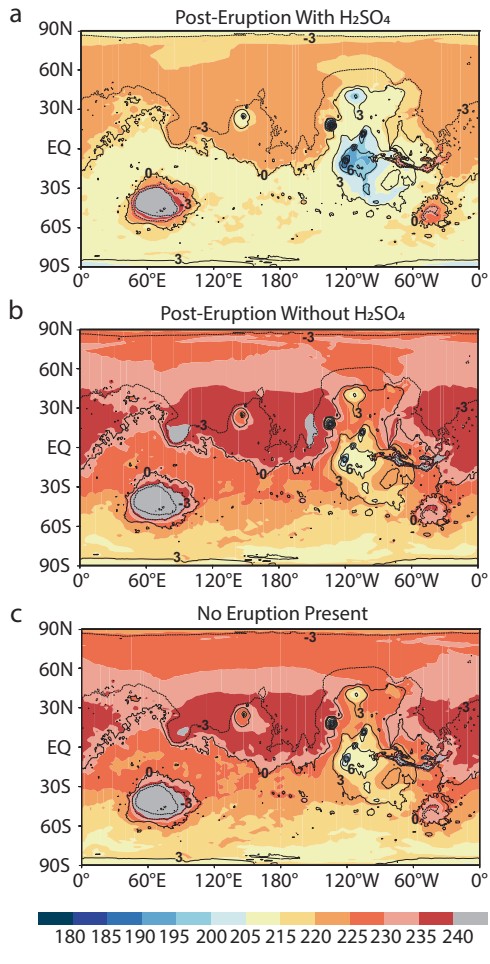

**Fig. 8 | Yearly averaged surface temperatures under three scenarios: with sulfuric acid, without sulfuric acid, and no eruption. a** The dispersal of sulfuric acid ($H_2SO_4$) in the year following an eruption from Apollinaris Mons on average leads to surface cooling across the planet compared to simulations with no $H_2SO_4$ present (**b**, **c**). The planetary average surface temperature decreases from 228 K in the control simulation (no eruption) to 217 K in the year following an eruption involving $H_2SO_4$. In contrast, eruptions without $H_2SO_4$ result in a planetary average surface temperature of 230 K, only slightly warmer than the control simulation. The eruption simulations are run under baseline conditions as listed in Table 2, except with a water mass eruption rate of $10^7$ kg s$^{-1}$. In the case with $H_2SO_4$ present, the $H_2SO_4$ mass eruption rate is $10^5$ kg s$^{-1}$. Modeled temperatures are overlaid onto a cylindrical projection of Mars. Black lines represent MOLA topographic contours, with numbers indicating contour elevations in kilometers.

and Syrtis Major source vent (i.e., 174.4°E, −8.5°S and 67.17°E, 9°N, respectively). The model then distributes this MER (kg s$^{-1}$) into the mass of air in a given grid box (kg), resulting in a kg kg$^{-1}$ s$^{-1}$ volcanic tendency:

$$V = \frac{F_{MER}g}{dPA} \qquad (2)$$

When a volcano erupts explosively, the resulting airborne ash particles are bound together via electrostatic attraction, moist adhesion with other particles, and hydrometeor formation (i.e., rain, graupel, hail, snow, etc.)[27] and references therein. We do not directly model this process of ash-water or ash-sulfur aggregation (aggregation processes will be handled in future work); instead we separately simulate the thermal and physical effects of ash aggregation with water by

varying the ice albedo and the number of CCN per unit mass of air in sensitivity tests as free parameters (discussed further in "Environmental parameters"). We assume that water vapor from the eruption freezes into ice via heterogenous nucleation with a fixed amount of cloud condensation nuclei (i.e., ash or dust) and eventually settles to the surface as an ash-ice mixture, which affects the ice albedo and consequently the rate of sublimation. We expect that modeling the details of aggregation would change the extent and thickness of the deposit by producing ice distributions that are closer to the source volcano due to the greater settling rate of ash-ice, ash-sulfur, or ice-sulfur aggregates relative to ice alone[28,93].

Volcanic $CO_2$ and $SO_2$ gases are expected to have some climatic impact[55,89,91,94,95] which may be further impacted by the scavenging of sulfur by particles of ash[96,97]. The effects of these gases on the climate is handled in a separate study[94] and the scavenging of sulfur particles by ash will be subject of future work, though we do explore the influence of sulfate aerosols on surface ice accumulation (discussed further in "Sulfuric acid emissions" and "Eruption parameters").

## Water precipitation and deposition

The release of water from explosive eruptions is handled by a self-consistent, integrated representation of the water cycle included in the PCM, taking into account atmosphere–surface interactions, transport, and the radiative effects of $H_2O$ gas and clouds[14]. Upon release into the atmosphere from an eruption, water vapor condenses to form clouds made of liquid water droplets or water ice particles, depending on the atmospheric temperature and pressure. In the PCM, erupted water is represented as spherical particles that are advected by winds and settle to the surface based on their calculated settling velocities. In reality, non-spherical particles may fall more slowly than spherical ones of the same radius, potentially resulting in longer atmospheric residence times than estimated here[98]. Local $H_2O$ cloud particle radii are based on the concentration of condensed material and are determined by the amount of activated cloud condensation nuclei per unit mass of air, $N_C$ (i.e., when water vapor in the air condenses onto these particles, they grow into cloud droplets, effectively "activating" the CCNs). $N_C$ is taken to be constant everywhere in the atmosphere (but varied in sensitivity tests; discussed further in "Environmental parameters") and is equal to $10^5$ kg$^{-1}$ (particles per kg of air) for both water ice and vapor clouds in the baseline simulation. Following the procedure outlined in previous PCM studies[14,21,89,99,100], the effective radius of the cloud particles is given by:

$$r = \left(\frac{3q_c}{4\pi\rho_c N_C}\right)^{\frac{1}{3}} \qquad (3)$$

Where $q_c$ is the mass mixing ratio of cloud particles (in kg kg$^{-1}$ of air), and $\rho_c$ is the density of the cloud particles (1000 kg m$^{-3}$ for liquid water and 920 kg m$^{-3}$ for water ice, corresponding to hexagonal ice—the most common crystalline form on Earth, and expected under early Mars conditions). The effective radii of cloud particles are then used to calculate their settling velocities and, following Mie theory—which describes light scattering by spherical particles of comparable size to the wavelength—to compute their radiative properties. Mie scattering is incorporated by generating lookup tables of the extinction coefficient (the strength of absorption or reflection at a given wavelength), single-scattering albedo (the ratio of scattered to absorbed light), and the Henyey-Greenstein asymmetry factor (the degree of forward or backward scattering). These tables are then used by the PCM to compute cloud scattering parameters[101].

The precipitation of water is handled by dividing precipitation into liquid and ice, depending on the temperature of the cloud particles. The precipitation of water is added to the total surface water loading (expressed in kg m$^{-2}$ and, in the case of ice deposition, can be converted to meters by dividing by the density of ice, which is

920 kg m$^{-3}$) at a precipitation rate (in kg m$^{-2}$ s$^{-1}$) calculated using the settling velocity of particles assumed to be equal to the terminal fall velocity that we approximate by Stokes law, with a "slip-flow" correction[102]. In the absence of precipitation, frost (i.e., ice deposition when the atmospheric relative humidity is high) may accumulate and contribute to the total loading of surface ice.

### Sulfuric acid emissions

During some terrestrial explosive volcanic eruptions, sulfur dioxide ($SO_2$) is injected into the stratosphere, where it reacts with water from the eruptive plume or from external water present in the environment to form sulfuric acid ($H_2SO_4$) aerosols[53–55,96,97]. The resultant stratospheric aerosol cloud has the ability to reflect solar radiation back into space, making it one of the major contributors to global cooling on Earth[54,55]. The aftermath from the presence of $H_2SO_4$ aerosols in the atmosphere may have also led to cooling on Mars[91,94,103], which could potentially influence the rate of ice accumulation following an eruption.

To investigate the effect of volcanic $H_2SO_4$ aerosols on surface ice accumulation, we follow the approach of Kerber et al. (2015), implementing a simplified scheme to assess the sensitivity of our results to $H_2SO_4$-induced cooling without explicitly modeling the formation process. Using Mie theory (as described in "Water precipitation and deposition"), we generate lookup tables for the PCM to calculate cloud scattering parameters of $H_2SO_4$ in each grid box[101]. To simulate emissions of $H_2SO_4$ that might be generated during an explosive event, we treat its MER as a free parameter and inject it into the model as physically and radiatively active, but chemically inert 0.3-μm spherical tracers (particle size is based on typical terrestrial volcanic aerosol radii[91]). The value of the $H_2SO_4$ MER is discussed further in "Eruption parameters". These tracers are transported by winds and settle gravitationally according to calculated settling velocities, without undergoing condensation. Furthermore, we treat $H_2SO_4$ as a homogenous liquid droplet aerosol and do not directly model the process of heterogeneous nucleation in which $H_2SO_4$ coats volcanic ash in the atmosphere. Modeling the photochemical formation of $H_2SO_4$ from $SO_2$ and its microphysical interactions with $H_2O$ in clouds and ash for the purpose of understanding its climatic effect will be the subject of future work.

### Fixed parameters

Table 1 lists our fixed input parameters used in this study. Before releasing explosively derived water vapor into the PCM, we set all other water reservoirs to zero and run a dry, one-year control simulation before each explosive eruption so that the climate may adjust to orbital and atmospheric changes[14]. Previous work simulating volcanic degassing into a wet atmosphere (i.e., with polar water ice caps present)[21] revealed negligible differences in the overall distribution of volcanically derived ice. We therefore move forward with assuming a dry atmosphere in our simulations, since this approach allows us to clearly track the behavior of volcanically derived water and observe the sensitivity of its distribution to our chosen parameters.

Previous studies suggest that early Mars had a mean surface pressure of at least 1 bar, which later declined rapidly to its present state[104–108]. Additionally, Mars may have transitioned from a reducing (hydrogen-rich) to an oxidizing (oxygen-rich) atmosphere during the early Noachian era[109], with brief episodes of reducing conditions triggered by impactor events, volcanism, crustal alteration, and atmospheric escape[110]. In this study, we assume a mean surface pressure of 1 bar $CO_2$, consistent with previous models of the early Martian atmosphere and similar to prior work using the Generic PCM to simulate early Martian atmospheric conditions[14,89,94]. To represent the oldest caldera ages of Apollinaris Mons and Syrtis Major, which are approximately 3.7–3.9 billion years old[42,43], we employ a consistent stellar flux (i.e., the amount of solar energy per unit area) value of

1024.5 W m$^{-2}$, that corresponds to solar spectral irradiance estimates from that period[111].

Mars' eccentricity is fixed at a long-term averaged value of 0.069[18] to account for variances in Mars' orbit throughout the lifetime of the volcanoes used in this study. The longitude of perihelion is fixed at its present-day value of $L_S = 251°$ (i.e., late northern fall) since the longitude of perihelion is expected to have repeatedly switched between the present and reverse orientation on a 50 ka cycle[8].

We use a parameterized version of the dust cycle that assumes a uniform visible optical depth of 0.2[112] in the atmosphere and a well-mixed vertical distribution up to 30 km. The dust radiative properties and scattering parameters are then evaluated in the infrared and visible wavelengths and are computed by the PCM in each grid box. This parameterization allows us to simulate the thermal effects of dust without the use of a dust transport scheme that determines the fraction of dust particles involved in cloud formation[99]. Instead, the impacts of dust on the number of available CCN for water cloud formation is handled separately by modifying the CCN parameter directly as a free parameter (discussed in "Environmental parameters").

Detailed sensitivity analyses of our fixed environmental parameters, including the presence of polar ice caps, mean surface pressure, eccentricity, longitude of perihelion, and dust visible optical depth, with respect to volcanic water outgassing are presented in our previous study[21].

### Sensitivity studies

We proceed with several sensitivity studies to test the rate of precipitation and surface ice accumulation from explosive volcanic eruptions on early Mars using Apollinaris Mons as our baseline volcano. Table 1 lists model parameters used for our sensitivity studies. Table 2 lists our baseline parameters, which include a water MER of $10^9$ kg s$^{-1}$, a plume height of 45 km, eruption duration of 3 sols beginning at the start of northern spring ($L_S = 0°$), an obliquity of 37.62°, an ice albedo of 0.645, the number of cloud condensation nuclei set to $10^5$ kg$^{-1}$ for both liquid water and ice clouds, and a post-Tharsis bulge topography. Each parameter is then individually varied over a range as described in "Eruption Parameters", "Environmental Parameters", and in Table 1 to test its sensitivity.

### Eruption parameters

The MER describes the intensity of an explosive eruption. Total (i.e., ash+gas) MERs for explosive terrestrial silicic and andesitic eruptions typically range from $10^6$ to $10^8$ kg s$^{-1}$, whereas explosive basaltic eruptions typically range from $10^3$ to $10^6$ kg s$^{-1}$ [64], although rates of up to $10^8$ kg s$^{-1}$ have been estimated for basalts[113]. Silicic and andesitic magmas result in higher MERs, not only because they tend to have higher total volatile contents, but also because the high silica content of the magma results in high viscosity magmas that entrap volcanic gases and promote high-explosivity eruptions. Basaltic magmas, however, tend to have lower volatile concentrations and have lower silica contents and thus, have lower viscosities, which may allow for more effective passive gas escape, the combined effect resulting in lower MERs. Martian volcanic materials consist predominantly of low-silica basalts, reflecting a general scarcity of more evolved lava compositions[114,115]. While basaltic magmas typically erupt effusively or with low explosivity on Earth, theoretical studies suggest that early Martian eruptions may have been more explosive due to the planet's lower gravity (3.7 m s$^{-2}$ on Mars versus 9.8 m s$^{-2}$ on Earth)[24,73,115]. Lower gravity on Mars reduces lithostatic pressure at depth, facilitating volatile exsolution, bubble nucleation, and magma disruption deeper within the subsurface. Consequently, microlite nucleation may also happen earlier, significantly increasing magma viscosity[116]. Early bubble nucleation provides more time for gas expansion during ascent, increasing gas volume and buoyancy, which accelerates magma rise.

As the magma ascends rapidly, gas expands violently when near-surface pressure drops, leading to fragmentation and potentially higher MERs. Additionally, lower gravity promotes taller eruption columns that carry particles to greater vertical and lateral distances, creating the potential for ash and ice to cover greater areas. The presence of abundant surface water on early Mars could further enhance explosivity when magma interacts with external water. As a result, explosive basaltic eruptions on early Mars may have resembled terrestrial Plinian eruptions, reaching heights of tens of kilometers[117,118]. As Mars transitioned into the Hesperian and atmospheric pressure declined[108], the thinning atmosphere—combined with the planet's low gravity—may have further amplified the explosivity of volcanic eruptions, potentially driving plumes into an inertial regime, behaving more like jets in which pyroclasts travel ballistically, governed primarily by inertia and gravity[119].

Studies of eruption characteristics on Mars have primarily focused on a range of MERs analogous to what we might expect in terrestrial eruptions (i.e., total MERs of $10^6$–$10^9$ kg s$^{-1}$[118,120,121]). However, considering Mars' lower gravity may permit eruption rates to be higher than their terrestrial counterparts, we test higher total mass eruptions rates of $10^8$, $10^9$, $10^{10}$, and $10^{11}$ kg s$^{-1}$, which correspond to water MERs of $10^6$, $10^7$, $10^8$, and $10^9$ kg s$^{-1}$ assuming a water content of 1 wt% for Martian magmas[122]. Total eruption rates of $10^8$ and $10^9$ kg s$^{-1}$ are akin to the 1980 Mount St. Helens eruption and 1991 Pinatubo eruption, respectively. Although higher total MERs of $10^{10}$ and $10^{11}$ have not been observed on Earth, we include these rates to account for the higher MERs that may have occurred on Mars. We assume a baseline water MER of $10^9$ kg s$^{-1}$ to gain insight on the maximum amount of water that can accumulate during an eruptive event. When testing the effects of $H_2SO_4$, we use a fixed $H_2SO_4$ MER of $10^5$ kg s$^{-1}$ and a water MER of $10^7$ kg s$^{-1}$ based on estimates from the Pinatubo June 15–16th, 1991 eruption assuming half of the $SO_2$ released was converted into $H_2SO_4$[53,67]. The atmospheric chemistry of Mars may influence the efficiency of this conversion. Previous work indicates that if early Mars had an oxidizing atmosphere (e.g., predominantly $CO_2$), $H_2SO_4$ readily forms from volcanic $SO_2$ outgassing, while more reducing conditions may have favored the formation of $S_8$[94]. In our model we assume an oxidizing atmosphere and the conversion of $SO_2$ to $H_2SO_4$, consistent with the transition from a reducing to an oxidizing atmosphere early in the Noachian era[109].

The eruption duration in this study is described as the time period in which water and $H_2SO_4$ are continuously emitted into the Martian atmosphere and is primarily dictated by the total supply of magma and dissolved volatiles. Individual terrestrial explosive eruptions can last as long as one day[64]. In contrast, fluid convective motions and diapiric ascent rates within the mantle are expected to be slower on Mars due to its lower gravity. Consequently, to prevent premature cooling and solidification, magma diapirs on Mars need to be larger than those on Earth to reach shallow crustal levels[73]. This requirement for larger diapirs at shallower depths might lead to explosive eruptions lasting for extended periods. Martian lavas can provide additional clues for the duration of volcanic activity. For example, the presence of lava flows and volumes larger than terrestrial lava flows by up to an order of magnitude[73,123] also suggests a greater availability of magma compared to Earth and hence, longer eruption durations. To capture a wide range of scenarios for the duration of explosive eruptions on Mars we test durations of 1, 3, and 5 sols.

Plume height estimates given in most terrestrial and planetary applications are based on conservation relationships established by Morton et al., (1956)[58] for buoyant convective rise. As described by the Morton-type convective rise formulation, the plume height, and thus, the neutral buoyancy height and release height of water and $H_2SO_4$, is ultimately controlled by the MER. Mars adaptations of this simple first-order eruption plume model suggest that for a total MER of $10^8$ kg s$^{-1}$, plume heights should range between 32 and 37 km for an early

atmosphere[118,121,124]. For a total MER of $10^9$ kg s$^{-1}$, theoretical assessments of Plinian eruptions suggest that corresponding plume heights should reach a maximum of 65 km[120] (Supplementary Table 2). Determining plume heights for higher total MERs not covered in Martian plume models (i.e., $10^{10}$ and $10^{11}$ kg s$^{-1}$) is not as straightforward.

The buoyancy of an eruptive plume depends on the density contrast between the eruptive cloud and the ambient atmosphere[125,126]. Models using Morton-type convective rise formulations suggest that for terrestrial eruptions, total MERs >$10^9$ kg s$^{-1}$ result in the convective plume becoming denser than the surrounding atmosphere, the collapse of the eruptive column, and the formation of pyroclastic density currents, which are ground-hugging mixtures of hot gas and ash that form deposits called ignimbrites[73,125,127,128].

However, it is uncertain if column collapse would also occur under similar conditions on early Mars, given the reduced gravity and uncertainties in the atmospheric density and temperature profile. The atmospheric temperature of early Mars remains debated and may have been colder than Earth, even at similar atmospheric pressures, due to Mars receiving less solar energy because of its greater orbital distance and the faintness of the young Sun[129]. Magma erupting into colder air experiences a greater temperature contrast compared to magma erupting into warmer air, which, all else being equal, can drive convecting Plinian columns to higher altitudes[130]. Moreover, the planet's lower gravity leads to higher columns and makes column collapse less likely for the same eruption energy and particle load. Even if column collapse and ignimbrite formation does occur, water present in the column would still get released into the atmosphere and may contribute to surface ice deposits. Moreover, collapsing columns often generate co-ignimbrite plumes, which form above pyroclastic density currents after they sediment coarse clasts, leaving fine particles, volcanic gases, and heated entrained ambient air to lift-off and ascend buoyantly. These plumes can rise nearly as high as the main eruption column, as documented in notable eruptions, such as Toba (75 ka), Tambora (1815), Mount St. Helens (1980), Pinatubo (1991), and the Campanian Ignimbrite eruption (~40 ka)[131–134]. In some cases, co-ignimbrite plumes may persist for longer durations and introduce more tephra and volatiles into the atmosphere than in the initial Plinian phase, with a MER comparable to that of the Plinian stage[134]. Our modeling approach accounts for this possibility by assuming that most of the material from a co-ignimbrite cloud (if it occurs) is injected into the atmosphere with a MER and column height similar to a Plinian eruption. However, if a collapsed column deposits substantial ice near the vent, our model may underestimate proximal ice deposition.

Given the uncertainty surrounding the atmospheric profile of early Mars, the plume height reached for a given MER is also not well constrained. Although this issue and column collapse dynamics warrant further study, here we opt to decouple MER and plume height to explore their effects on ice distribution independently. We conduct several additional runs to test plume heights of 35, 45, and 65 km for each MER.

The injection of water into the atmosphere at these plume heights may expose the water vapor to winds capable of transporting it to even higher altitudes, potentially even into outer space. Escape processes, such as Jeans escape, which may explain the loss of hydrogen from Mars, occur at altitudes between 100 and 250 km and operate on timescales of days[135,136]. Since we do not model atmospheric escape in this study, our surface ice distributions may represent the maximum potential contribution from explosive eruptions. However, much of this water may have precipitated out of the atmosphere on timescales shorter than the escape rate. Alternatively, it could have co-deposited with volcanic ash, preventing it from reaching such high altitudes. In either case, the water may have become trapped beneath the surface or chemically incorporated into surface materials[137–140], thereby

protecting it from sublimation and subsequent escape. The influence that atmospheric escape processes have on tall eruptive plumes remains uncertain and more work on this subject is warranted.

## Environmental parameters

In our baseline simulations, we use present-day surface topography derived from the Mars Orbiter Laser Altimeter (MOLA) onboard the Mars Global Surveyor[141,142], as most of the largest topographic features, including the Tharsis bulge, hemispheric dichotomy boundary, and the Hellas basin, likely pre-date the formation of Apollinaris Mons and Syrtis Major[143–145]. However, given the potential overlap between volcanic activity and the formation of the Tharsis bulge[42], we evaluate the sensitivity of surface ice accumulation to pre-Tharsis topography, as the bulge's high elevation could have acted as a trap for water ice and significantly influenced its spatial distribution.

Cloud condensation nuclei play a crucial role in cloud formation through heterogeneous nucleation. In this process, impurities, such as volcanic ash, assist in crystallization by lowering the energy barrier for condensation, requiring lower supersaturation, and producing more stable droplets[31,146]. During explosive eruptions, the abundance of fine ash particles can make volcanic clouds richer in ice-forming nuclei compared to typical meteorological clouds[31]. The number of CCN is initially taken to be constant throughout the atmosphere and is set $10^5$ particles per kg of air (kg$^{-1}$) for liquid and ice water clouds in our baseline simulation and represents a standard value used for studies of rocky planet climates[14,147]. We then perform a sensitivity study in which we test two additional values of the number of CCN to encapsulate its effects on cloud properties: $10^4$ kg$^{-1}$ and $10^6$ kg$^{-1}$ for both liquid and water ice clouds.

The ice albedo is an important parameter affecting the amount of solar energy surface ice absorbs. Small amounts of dust or ash can drastically alter the albedo of the ice and therefore affect how much solar energy the ice absorbs[148]. Varying the ice albedo allows us to parameterize the thermal effects of ice and ash that may have co-deposited to the surface (i.e., accretionary lapilli) by changing the propensity of ice to sublime. We test three different albedo scenarios: an ice albedo of 0.5 for dirty ice, 0.645 for moderately dirty ice, and 0.95 for fresh ice[14,147,149]. We assume an albedo of 0.645 as our baseline to represent a moderately lowered albedo due to the presence of dust and/or ash. We track the rate of surface ice sublimation using a test simulation that continues for one year following the cessation of an explosive eruption with a given ice albedo.

Previous work has suggested that changes in obliquity could explain excess hydrogen in equatorial regions[4]. To determine whether changes in obliquity are necessary to explain patterns of excess hydrogen, we perform a sensitivity test to assess how changes in insolation might influence volcanic ice precipitation and deposition. We test a range of obliquity values, including 0°, 25.19°, 37.62° (baseline), 45°, and 60°, to account for the orbital variations Mars may have experienced[18].

We test the sensitivity of ice deposition from explosive eruptions to changes in the season, which has been demonstrated to influence the directionality of volcanic water since the atmosphere is subject to changing seasonal winds[21,150]. We set explosive eruptions to run for 3 sols at the start of the northern spring (solar longitude (L$_S$) = 0°), summer (L$_S$ = 90°), fall (L$_S$ = 180°), and winter (L$_S$ = 270°). We further test how precipitation and surface ice accumulation respond to eruptions occurring at perihelion (L$_S$ = 251°)[151] and aphelion (L$_S$ = 71°)[151] to capture the insolation effects when Mars is nearest and furthest from the Sun.

## Data availability

The Generic PCM modified for this study's methodology and model input/output files are available at the following GitHub repository: https://github.com/sshamid1/Hamid_LMD_Generic_PCM_Volc and at https://doi.org/10.5281/zenodo.16891092.

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

## Acknowledgements

We thank Ehouarn Millour for his assistance with model set up and interpretations. This material is based upon work supported by the National Science Foundation Graduate Research Fellowship under Grant No. 026257- 001 to S.S.H. and NASA Solar System Workings Grant 20-SSW20-0086 to L.K.

## Author contributions

S.S.H. performed all climate model simulations, L.K. supervised the research, and A.B.C. supervised and advised on eruption parameters. All authors contributed to the interpretation of results and writing of the manuscript.

## Competing interests

The authors declare no competing interests.
