## [Transparent Peer Review file · Nature Communications]

Precipitation Induced by Explosive Volcanism on Mars and its Implications for Unexpected Equatorial Ice

Corresponding Author: Dr Saira Hamid

Version 0:

Reviewer comments:

Reviewer #1

(Remarks to the Author)

I have had the opportunity to review the manuscript "Precipitation Induced by Explosive Volcanism on Mars and its Implications for Ice in Unexpected Equatorial Regions." The study presents a compelling investigation into the potential link between ancient explosive volcanism and ice deposition on Mars. Notably, numerical model simulations suggest that water vapor released during explosive eruptions of Apollinaris Mons and Syrtis Major could have led to localized precipitation of ice and/or ice-ash aggregates. This finding offers a plausible explanation for the unexpected presence of ice near the Martian equator, contributing to a broader understanding of the planet's past climate dynamics, which is the subject of a concerted effort to understand.

I believe that the study is significant for the field of Martian science and holds relevance for related disciplines such as atmospheric modeling, volcanology, and astrobiology. While previous research has explored various hypotheses regarding equatorial ice on Mars, this work expands upon the existing literature by directly linking explosive volcanic activity in specific areas to potential ice deposition. The methodology appears robust, with well-justified input parameters and a clearly defined modeling approach. The results are convincingly presented and align with established knowledge of Martian atmospheric processes.

During my review, I did not identify major weaknesses in the argumentation or in the values used for the model simulations. Nor did I find significant textual issues requiring extensive revision. However, I suggest merging Section 3 into Section 2, as its single-paragraph length appears somewhat out of place. The brevity of the discussion is not problematic in itself—fragments of it are appropriately distributed throughout Sections 2 and 4—but its current structure is somewhat unconventional. Additionally, two minor improvements can be done - as making the red line in Figure 1 more visible and removing the redundant definition of the abbreviation CCN on line 517. Apart from these minor details, however, I find the manuscript prepared to a very good standard.

Overall, the manuscript is well-prepared, logically structured, and written in a clear and concise manner. The methods are described in sufficient detail to ensure reproducibility, making this study a valuable contribution to the ongoing discourse on Mars' climatic and geological history. Given these strengths, I recommend the article for publication.

Reviewer #2

(Remarks to the Author)

The manuscript "Precipitation Induced by Explosive Volcanism on Mars and its Implications for Ice in Unexpected Equatorial Regions" by Hamid et al. presents a novel way to produce precipitation and ice accumulation at the equatorial region, as well as a mechanism for preserving this water ice which may explain the observed high hydrogen content at certain equatorial regions. I would like to commend the authors on this work. Not only have they presented a novel idea regarding ice accumulation on past Mars and preservation of ice, the science presented is well done and comprehensive. Furthermore, this paper is well written and clearly presented. I have no major comments or concerns regarding the writing or content of this manuscript. I only offer one comment / thought for the authors consideration. I recommend that this manuscript be accepted in Nature Communications as it has a high impact for the Mars science community.

A comment / thought: I fully understand linking explosive volcanism induced precipitation to the low latitude measurements

of hydrogen given that these appear to correlate with features that may have been large-scale past explosive volcanic eruptions. I do wonder if the authors have thought about this mechanism as a way to deposit ice/snow during the Early Hesperian valley network period (and potentially crater lakes) or later-stage hydrologic activity related to the deposition of alluvial fans (Hesperian to Early Amazonian). Given the older ages of these volcanic epicenters, the distribution of ice predicted by the models, and the potential to melt this ice and/or snow during periods of varying obliquity, I think this has some potential very interesting applications to past epochs of aqueous activity on Mars and if possible I would be interested in some discussion on this mechanism related to those periods on Mars. I do understand that space is limited in a paper such as this, but it would be interesting to see more discussion on this as a source for other aqueous periods on Mars (particularly with relation as a possible snow/ice depositional mechanism for a freeze thaw source for the alluvial fans).

Reviewer #3

(Remarks to the Author)

Review of "Precipitation Induced by Explosive Volcanism on Mars and its Implications for Ice in Unexpected Equatorial Regions"

Broad Thoughts:

This manuscript explores the effects of explosive volcanism as a significant source for the anomalously high amounts of water found in equatorial regions of Mars as measured by remote sensing satellites. The authors use a GCM to simulate ash and water dispersal and explore myriad parameters that could affect the resulting ash/water distribution patterns and longevity, including obliquity, eruption rate, and sulfur content (amongst others). The authors find that significantly large explosive eruptions can produce meters of water ice in the areas that are observed to be enriched in hydrogen.

Overall, the manuscript makes some convincing arguments, and the results are really neat, however, there are some moderately significant issues that need to be corrected before publication. I've highlighted some of the main problems below. With the correction of these issues, this manuscript would be an excellent addition to the literature.

1. Regarding article scope, I don't specifically call this out anywhere, but this article is in Nature Communications which reaches a broader audience than the planetary community. As such, the authors may want to revise some parts of the manuscript to be a bit friendlier to a non-planetary audience (one example comes to mind is not explaining what the hemispheric dichotomy is when it's first mentioned). Not a super big deal, but thought I'd mention it.
2. There are a several places in the manuscript where the authors cite other studies to back up claims or assumptions, but these studies deal with a modern day martian atmosphere, not an ancient one like is explored in this manuscript. This has some larger implications in that some of the assumptions the authors make are dependent on these incorrect citations which may affect their results. I've highlighted the ones I noticed in the text, but I worry there are likely more. I'd recommend going through all the citations and making sure they are for the correct era, as right now, the citations don't really mean anything. I also recommend revisiting the assumptions made throughout the manuscript and to make sure things still make sense in an ancient atmosphere.
3. The colorscale used in a lot of the figures (fig.1bcd,fig.2, fig.3) is misleading. In some places it seems to be logarithmic jumps between parts of the colorscale and others it seems to be linear. Whatever's going on, the differences between colors don't correspond to any regular pattern that I can see. Because of this, there will patterns on the map that will appear more important or focused than they truly are. The authors need to choose intervals that are either regularly spaced or on a logarithmic scale. I wouldn't be harping on this as hard as I am right now, but I think this is important here given the authors are relying on these figures to make their claims rather than having any statistical/quantitative analysis, especially when talking about agreements with excess hydrogen maps (lines 124-125)
4. How are the authors calculating ice thickness? This is a major finding that makes its way into the abstract, but I wasn't able to find anywhere in the manuscript where this is explained. The PCM calculates a mass loading, but to convert to a thickness requires the authors to assume a packing density. Given that both ash and snow/ice can have wildly different packing densities depending on all sorts of things (distance from the vent, presence of rain at deposition, crystal/clast size, etc.), this decision of density is somewhat critical to the author's results and should be backed up accordingly.
5. Some volcanological considerations regarding plume heights and mass eruption rates. I've bolded these in the line items (155-163 and 490-501). But in general, I have some concerns with how the authors are relating mass eruption rates and plume heights (especially those that aren't super likely) and how the authors are ignoring the effects of column collapse, especially in very large eruptions as a significant portion of ash would likely not be airborne in that case.

Line Items:

Line 15: Consider adding "which" after "regions" and replacing "containing" with "contain" to make it clearer that the authors are referring to existing measurements

General introduction thoughts: There's a lack of motivations in the intro besides asking what the origin of ice is. The authors could potentially lean into In Situ Resource Utilization (ISRU) as a means of appealing outside the planetary community (given the choice of journal)

Lines 26-38: It may serve to do a more extensive literature review with regards to the anomalous equatorial hydration signals, as there's a lot of hypotheses that aren't mentioned here that have significant scientific merit (e.g., hydrous alteration in an aqueous environment). Check out Karunatillake et al. (2014) or Hood et al. (2019) as some starting points.

Lines 28-30: How deep are these instruments sensitive to? It would be good to give that information somewhere. I know GRS is a couple decimeters, but I'm not sure what FRENDS's sensitivity is.

Line 36: Be careful with the term "permafrost" here. I believe it has a very specific definition on Earth that may not match what the authors are describing here. Calling it "bulk ice" or "ice-ash aggregate" may be better.

Lines 53-55: Hard to imagine an explosive eruption without ash. Consider removing "if accompanied by ash" and adding "also" after "may"

Line 58-59: More volatiles? Different volatiles?

Lines 59-70: All of the processes listed here don't usually get preserved in the terrestrial geologic record except in rare circumstances (e.g. raindrop imprints or fulgurites). The authors may need to more explicitly explain why they cite these examples.

Lines 97-98: Here and elsewhere, it would be great to refer readers to a context map to get a better sense of where all the features the authors have been mentioning are and which parts are hydrated.

Line 102: Consider rephrasing to: "While other workers(46) used volcanic ash deposition as a proxy for volcanic water enhancement in regolith, we explicitly model the release of water into the atmosphere as the result of an explosive eruption while tracking ice deposition"

Line 125: Where is this thickness coming from? How is it being calculated? I can't find anywhere in the methods that explains this. See main comment above.

Line 124-125: Was there any sort of statistical link found between the modeled results and the excess hydrogen data? Something beyond qualitative observations

Line 128: Climatologically analogous I assume

Lines 130-137 : Citation for these? Or figure callouts?

Figure 1 (and others): The colorscale on most of these figures is confusing to me. Is it logarithmic? Sometimes? It's misleading to make the color breaks at these inconsistent intervals as that may highlight certain features while downplaying others. I would suggest just making the colorscale fully logarithmic.

Also: Given the timescale listed for panel d, are we to assume panels b and c are immediately after eruptions? Please list that somewhere in the figure or in the figure caption.

Line 143 and elsewhere: Why are these figures in the supplemental if the concept is important enough to be a main section in the results section? Maybe some of these sections belong in a special model sensitivity section?

Lines 155-163: Can the authors also comment on the validity of really high MERs but low plume heights? Those don't tend to scale like that (see Mastin et al., 2009 and 2014)

Line 163: Why does the model crash if there's water at high altitude?

Lines 172-174: That's neat!

Line 199: There's an argument to be made here of very powerful yet sulfur-weak eruptions, such as the 2022 Hunga Tonga eruption which was largely phreatomagmatic and low ash and sulfur. Given that early Mars had a lot of water (Carr and Head III, 2003; Carr and Head, 2010; Scheller et al., 2021; Boynton et al., 2002; Squyres and Carr, 1986) for erupting magma to interact with, phreato eruptions likely occurred. Not saying the authors are wrong for including the effects of sulfur, just food for thought when making final conclusions of how much of a role sulfur played in ice deposition from eruptions.

Lines 251-255: The amount of ice being deposited is on the scale of a few meters right? Can these RADAR sounders resolve units that thin?

Figure 4: Consider adding some statistics to this figure (e.g. average, min/max, etc.) or to the text that references the figure. Also, a figure showing sublimation rate across the planet through time would be neat and help bring home the point about these rates being a dynamic thing.

Line 287: Consider replacing "code" with "calculations" and deleting "the" before "radiative"

Line 308-310: Did the authors perform any sensitivity tests with particles that are not spherical? If not, can they comment on

what effect nonspherical particles would have on their results given that ash is usually in quite unusual shapes?

Line 315: An explanation of what cloud condensation nuclei is/controls would be helpful here.

Line 319-321: Why? Is this a short process? A citation to a study of wet ash aggregation may be necessary. Van Eaton et al. (2015) and Textor et al. (2006ab) may be good places to start.

Line 338: CCN should be defined earlier.

Line 338: What does “activated” mean?

Line 339: It's never explained in section 4.5.2 how the authors calculate particle radii from CCN

Lines 350-352: It doesn't necessarily have to be water from the plume. It could also be environmental.

Line 362: What is a Mie code? What does that mean?

Line 366: Physically inert too?

Lines 376-380: Given the likely overlap of some of these volcano's ages and Tharsis's existence, it would be nice to see a sensitivity test of removing the Tharsis bulge (an easy knob to turn in the PCM)

Lines 428-434: Yes, but there's other things to consider here too such as magma ascent rate. There could also just simply be less magma to erupt due to a bunch of factors (local lithology, plate tectonic setting etc.)

Line 435-438: It's important to note here that both these citations give the “Mars has more explosivity” argument due to a current martian atmosphere, not an ancient one like is being simulated here. Consider removing this sentence or specify that it's for current martian pressure.

Lines 435-448: Also consider including the effects of available surface water or ice enhancing the explosivity of a basaltic eruption (Brož et al., 2021; Brož and Hauber, 2013; Sears et al., 2022; Wilson and Head III, 2004; Wilson and MouginisMark, 2003; Paladino et al. 2024)

Line 450: Think the authors lost a parenthesis here.

Line 461: Does this assumption hold up on Mars (those citations are for terrestrial eruptions)? Martian atmospheric chemistry is awfully different than terrestrial chemistry and the available constituent chemical ratios are likely different.

Line 482: I think this citation is wrong (should be Glaze & Baloga, 2002)

Lines 490-501: Ok, but the authors are simulating these eruptions in a 1 bar atmosphere which is fairly well established to be a reasonable estimate of martian surface pressure in the Noachian and identical in pressure to modern Earth so column collapse would most likely be occurring at these massive eruption rates. Collapse would also likely be occurring at mass eruption rates orders of magnitude lower than 10^{10} or 10^{11} kg/s. Given this and the fact that the PDCs tend to deposit significant amounts of material close to the vent (even if they form phoenix clouds), I'm not sure the authors can fully ignore the effects of column collapse.

I would suggest at least acknowledging this and providing some back-of-the-envelope calculations of how this would affect the authors' results or (if time and space allow) provide another scenario in which some degree of mass (20%? 30%?) is assumed to be deposited locally around the volcano and running sets of simulations at these reduced mass loads.

Line 511: Might be worthwhile to cite some ash hydration studies here (e.g. Aomine and Wada, 1962; Seligman et al., 2016; Hudak et al., 2021)

Lines 519-522: Similar to my comments on lines 435-438, the cited source here is for a study on a current martian atmosphere, not an ancient one. The authors might want to revisit their rationale in this section. In an ancient atmosphere, is heterogeneous nucleation still the dominant mechanism?

Line 529: Consider putting kg^{-1} after these numbers for consistency purposes.

Version 1:

Reviewer comments:

Reviewer #3

(Remarks to the Author)

2nd Review of, "Precipitation Induced by Explosive Volcanism on Mars and its Implications for Ice in Unexpected Equatorial Regions"

Broad Thoughts:

This manuscript explores the effects of explosive volcanism as a significant source for the anomalously high amounts of water found in equatorial regions of Mars as measured by remote sensing satellites. The authors use a GCM to simulate ash and water dispersal and explore myriad parameters that could affect the resulting ash/water distribution patterns and longevity, including obliquity, eruption rate, and sulfur content (amongst others). The authors find that significantly large explosive eruptions can produce meters of water ice in the areas that are observed to be enriched in hydrogen.

After the first round of review, the manuscript is looking in great shape. The authors addressed most comments made by myself and other reviewers in a well thought out and convincing manner, adding descriptions or new figures in the text where necessary. Overall, the study is well written, scientifically significant, convincing, and will make an excellent addition to the literature. A few more comments and line items to check out, but I don't need to see the manuscript again.

Great job, Saira! This looks like a ton of work and you've presented it well. Excited to see what you do next!

-Tyler Paladino

Comments:

Thank you for adding some information about the assumed density used for thickness calculations. I would still appreciate some commentary on why this value for density was chosen given it can change a lot with amount of ice or distance from vent.

For a bunch of the figures that have units of kg/m^2 , the label should read "surface ice mass loading" or something similar rather than "surface ice thickness"

Line items:

Figure 1: Love the context map. Could boxes be added to the panel a that correspond to the inset panels b and c to give an idea of where we're zoomed into?

Figure 3 (and others): Can the authors add a symbol to show where the eruptive vent(s) is on the map? Also, the numbers on the map are labeling the topo contours, right? That could be explicitly written out somewhere in the caption. Otherwise, I could see that causing some confusion.

Figure 5: The quiver plots are a bit incomprehensible right now. Consider removing maybe a quarter to a third of the quivers to make it a bit more obvious which way the wind is blowing. It might also be good to experiment with making all the quivers for each plot be on the same scale.

Lines 249-258: I was slightly confused when reading the discussion about grainsizes. The grain size the authors talk about is the ice/ash aggregate grainsize correct? Is this explicitly modeled or inferred? I didn't think the PCM could deal with actively changing grainsizes (but I might be wrong)

Lines 277-287: What timescale did the authors find for these protection effects to go away?

Lines 314-334: I'm still feeling a bit wary on this idea. I think it's ok to say that there were explosive eruptions as atmospheric pressure decreased, but the way the eruptions would behave would be different and wouldn't necessarily play by the rules or assumptions of normal plumes like are used in this study. I think just mentioning that somewhere in this section would be sufficient: That volcanoes in this low-pressure time periods may indeed have contributed to ice deposits, but that the way eruptions are modeled in this study is fundamentally different than how they would behave in a low-pressure atmosphere.

Lines 534-550: Thank you for addressing this. Might be worthwhile to tack on a bit to the end to acknowledge that plumes in a thinner atmosphere would behave pretty differently (see Wilson and Head, 2007 for their "inertial plumes" idea)

Lines 604-607: This needs a citation.

Lines 600-627: Thanks for adding this section! Adds some much needed discussion/caveats on the topic. Though I might push back a little on the "uncertainty of column collapse occurring in early Martian conditions" claim. Column collapse would definitely be happening at the modeled atmospheric pressures and gravity (there's a handful of geologic examples), but the degree to which it occurs within a single eruption is uncertain (i.e. how much material is collapsing). It's hard to have a plume that doesn't have some amount of material that's collapsing, especially at really high MERs.

Reviewer #1 (Remarks to the Author):

I have had the opportunity to review the manuscript “Precipitation Induced by Explosive Volcanism on Mars and its Implications for Ice in Unexpected Equatorial Regions.” The study presents a compelling investigation into the potential link between ancient explosive volcanism and ice deposition on Mars. Notably, numerical model simulations suggest that water vapor released during explosive eruptions of Apollinaris Mons and Syrtis Major could have led to localized precipitation of ice and/or ice-ash aggregates. This finding offers a plausible explanation for the unexpected presence of ice near the Martian equator, contributing to a broader understanding of the planet’s past climate dynamics, which is the subject of a concerted effort to understand.

I believe that the study is significant for the field of Martian science and holds relevance for related disciplines such as atmospheric modeling, volcanology, and astrobiology. While previous research has explored various hypotheses regarding equatorial ice on Mars, this work expands upon the existing literature by directly linking explosive volcanic activity in specific areas to potential ice deposition. The methodology appears robust, with well-justified input parameters and a clearly defined modeling approach. The results are convincingly presented and align with established knowledge of Martian atmospheric processes.

During my review, I did not identify major weaknesses in the argumentation or in the values used for the model simulations. Nor did I find significant textual issues requiring extensive revision. However, I suggest merging Section 3 into Section 2, as its single-paragraph length appears somewhat out of place. The brevity of the discussion is not problematic in itself—fragments of it are appropriately distributed throughout Sections 2 and 4—but its current structure is somewhat unconventional. Additionally, two minor improvements can be done - as making the red line in Figure 1 more visible and removing the redundant definition of the abbreviation CCN on line 517. Apart from these minor details, however, I find the manuscript prepared to a very good standard.

Overall, the manuscript is well-prepared, logically structured, and written in a clear and concise manner. The methods are described in sufficient detail to ensure reproducibility, making this study a valuable contribution to the ongoing discourse on Mars’ climatic and geological history. Given these strengths, I recommend the article for publication.

Thank you for your feedback! We have incorporated your suggestions of combining section 3 into section 2, improving the visibility of Figure 1, and removing the redundant definition of CCN in Section 3.3.2 into the paper.

Reviewer #2 (Remarks to the Author):

The manuscript "Precipitation Induced by Explosive Volcanism on Mars and its Implications for Ice in Unexpected Equatorial Regions" by Hamid et al. presents a novel way to produce precipitation and ice accumulation at the equatorial region, as well as a mechanism for preserving this water ice which may explain the observed high hydrogen content at certain equatorial regions. I would like to commend the authors on this work. Not only have they presented a novel idea regarding ice accumulation on past Mars and preservation of ice, the science presented is well done and comprehensive. Furthermore, this paper is well written and clearly presented. I have no major comments or concerns regarding the writing or content of this manuscript. I only offer one comment / thought for the authors consideration. I recommend that this manuscript be accepted in Nature Communications as it has a high impact for the Mars science community.

A comment / thought: I fully understand linking explosive volcanism induced precipitation to the low latitude measurements of hydrogen given that these appear to correlate with features that may have been large-scale past explosive volcanic eruptions. I do wonder if the authors have thought about this mechanism as a way to deposit ice/snow during the Early Hesperian valley network period (and potentially crater lakes) or later-stage hydrologic activity related to the deposition of alluvial fans (Hesperian to Early Amazonian). Given the older ages of these volcanic epicenters, the distribution of ice predicted by the models, and the potential to melt this ice and/or snow during periods of varying obliquity, I think this has some potential very interesting applications to past epochs of aqueous activity on Mars and if possible I would be interested in some discussion on this mechanism related to those periods on Mars. I do understand that space is limited in a paper such as this, but it would be interesting to see more discussion on this as a source for other aqueous periods on Mars (particularly with relation as a possible snow/ice depositional mechanism for a freeze thaw source for the alluvial fans).

Thank you very much for your supportive feedback! :D

We have incorporated your suggestion into Section 2.4. The second and third paragraph of this section now reads:

The shift from explosive volcanic activity in early Mars to effusive activity later in Mars' history coincides with the decline in atmospheric pressure, a potential transition from a wetter to drier mantle, and the loss of surface water reservoirs (Brož et al., 2021). However, low atmospheric pressure may still facilitate sporadic explosive eruptions. Reduced pressure allows for greater gas expansion and more efficient energy release, resulting in higher eruption velocities and more thorough magma fragmentation, leading to increased explosivity even with lower volatile concentrations (Wilson and Head, 1994). Eruptions under a thinner atmosphere may have promoted more surface ice deposition due to the reduced capacity of the atmosphere to retain moisture. Additionally, the faster winds associated with the thinner atmosphere could have enabled broader dispersal of volcanic water (Hamid et al., 2024). We thus propose that younger shield volcanoes on Mars may have undergone episodic explosive activity, potentially contributing to ice-rich substrates and morphologies on both local and global scales. Potential examples include Elysium Mons (Christiansen, 1985; Russell and Head, 2003), Hecates Tholus (Hauber et al., 2005; Fassett et al., 2006), Olympus Mons (Head et al., 2005), Alba Mons (Schiff

and Gregg, 2022), Arsia Mons (Head and Marchant, 2003), and the low shields within the Tharsis volcanic province (Pieterik and Jones, 2024).

Eruptions occurring throughout Mars' history, including under reduced atmospheric pressure, may have contributed to past aqueous activity. This includes the valley network period (and possibly paleolakes) formed around the Noachian-Hesperian boundary (~3.8–3.6 Ga; Hynek et al., 2010), minor valley formation continuing through the Hesperian and into the Early Amazonian (~2.8 Ga; Hynek et al., 2010), and hydrologic events linked to the formation of alluvial fans from the Hesperian to the Amazonian (~3.7–2.5 Ga; Holo et al., 2021). In essence, explosive volcanic eruptions may have dramatically transformed Mars' surface over time, redistributing volatile resources without requiring changes in planetary obliquity.

Reviewer #3 (Remarks to the Author):

Review of “Precipitation Induced by Explosive Volcanism on Mars and its Implications for Ice in Unexpected Equatorial Regions”

Broad Thoughts:

This manuscript explores the effects of explosive volcanism as a significant source for the anomalously high amounts of water found in equatorial regions of Mars as measured by remote sensing satellites. The authors use a GCM to simulate ash and water dispersal and explore myriad parameters that could affect the resulting ash/water distribution patterns and longevity, including obliquity, eruption rate, and sulfur content (amongst others). The authors find that significantly large explosive eruptions can produce meters of water ice in the areas that are observed to be enriched in hydrogen.

Overall, the manuscript makes some convincing arguments, and the results are really neat, however, there are some moderately significant issues that need to be corrected before publication. I've highlighted some of the main problems below. With the correction of these issues, this manuscript would be an excellent addition to the literature.

1. Regarding article scope, I don't specifically call this out anywhere, but this article is in Nature Communications which reaches a broader audience than the planetary community. As such, the authors may want to revise some parts of the manuscript to be a bit friendlier to a non-planetary audience (one example comes to mind is not explaining what the hemispheric dichotomy is when it's first mentioned). Not a super big deal, but thought I'd mention it. *In lines 88–91 we now define the Martian hemispheric dichotomy when it is first mentioned. The sentence now reads: Meridiani Planum is a plain located at 0.04°S, 3.14°W near the hemispheric dichotomy boundary (i.e., topographic division in elevation and crustal thickness between the Northern Lowlands and Southern Highlands of Mars).*

We also better explain some of the terminology we use. Specifically, in lines 221–224 when we use the term zonal winds we add clarification for the meaning of this term: Using Apollinaris Mons as an example case, we find that plumes reaching higher altitudes are subjected to

stronger zonal winds (winds that flow along lines of latitude) (Fig. 5), resulting in water being transported through the atmosphere for extended periods.

2. There are a several places in the manuscript where the authors cite other studies to back up claims or assumptions, but these studies deal with a modern day martian atmosphere, not an ancient one like is explored in this manuscript. This has some larger implications in that some of the assumptions the authors make are dependent on these incorrect citations which may affect their results. I've highlighted the ones I noticed in the text, but I worry there are likely more. I'd recommend going through all the citations and making sure they are for the correct era, as right now, the citations don't really mean anything. I also recommend revisiting the assumptions made throughout the manuscript and to make sure things still make sense in an ancient atmosphere. *There are two primary cases that were highlighted and are related to heterogeneous nucleation and plume behavior in the early Martian environment.*

Heterogeneous nucleation of clouds in the atmosphere: *Heterogeneous nucleation is always energetically favored over homogeneous nucleation for cloud formation. Assuming that the ancient Martian atmosphere had particles in it (e.g., ash or dust), heterogeneous nucleation would be the primary mechanism of cloud formation, whether the atmosphere was ancient or modern.*

We did clarify some text. Section 3.3.2 lines 651–654 now reads: Cloud condensation nuclei play a crucial role in cloud formation through heterogeneous nucleation. In this process, impurities, such as volcanic ash, assist in crystallization by lowering the energy barrier for condensation, requiring lower supersaturation, and producing more stable droplets (Durant et al., 2008, Dunning et al., 1960). During explosive eruptions, the abundance of fine ash particles can make volcanic clouds richer in ice-forming nuclei compared to typical meteorological clouds (Durant et al., 2008).

Plume behavior in the early Martian atmosphere — effects of lower gravity: *We base our assumptions on the Wilson and Head (1994) study which includes theoretical assessments for how a volcanic plume would behave in the lower atmospheric pressure of Mars in addition to the lower gravity.*

We added clarification in the text in lines 534–550 that we are citing these sources to appeal to the argument of low gravity (and not low atmospheric pressure) in facilitating higher explosivity eruptions:

While basaltic magmas typically erupt effusively or with low explosivity on Earth, theoretical studies suggest that early Martian eruptions may have been more explosive due to the planet's lower gravity (3.7 m s^{-2} on Mars versus 9.8 m s^{-2} on Earth) (Brož et al., 2021; Grott et al., 2013; Wilson and Head, 1994). Lower gravity on Mars reduces lithostatic pressure at depth, facilitating volatile exsolution, bubble nucleation, and magma disruption deeper within the subsurface. Consequently, microlite nucleation may also happen earlier, significantly increasing magma viscosity (Wall et al., 2014). Early bubble nucleation provides more time for gas expansion

during ascent, increasing gas volume and buoyancy, which accelerates magma rise. As the magma ascends rapidly, gas expands violently when near-surface pressure drops, leading to fragmentation and potentially higher mass eruption rates. Additionally, lower gravity promotes taller eruption columns that carry particles to greater vertical and lateral distances, creating the potential for ash and ice to cover greater areas. The presence of abundant surface water on early Mars could further enhance explosivity when magma interacts with external water. As a result, explosive basaltic eruptions on early Mars may have resembled terrestrial Plinian eruptions, reaching heights of tens of kilometers (Woods, 1995; Hort and Weitz, 2001). As Mars transitioned into the Hesperian and atmospheric pressure declined (Scherf and Lammer, 2021), the thinning atmosphere, combined with the planet's low gravity, may have further amplified the explosivity of volcanic eruptions.

3. The colorscale used in a lot of the figures (fig.1bcd,fig.2, fig.3) is misleading. In some places it seems to be logarithmic jumps between parts of the colorscale and others it seems to be linear. Whatever's going on, the differences between colors don't correspond to any regular pattern that I can see. Because of this, there will patterns on the map that will appear more important or focused than they truly are. The authors need to choose intervals that are either regularly spaced or on a logarithmic scale. I wouldn't be harping on this as hard as I am right now, but I think this is important here given the authors are relying on these figures to make their claims rather than having any statistical/quantitative analysis, especially when talking about agreements with excess hydrogen maps (lines 124-125). *We have updated figures 2–7 so that the modeled distribution of ice are in logarithmic scales. Using log scale for our modeled distribution doesn't affect the agreement with excess hydrogen maps, mostly just the colors that are associated with different contour levels of the ice.*

4. How are the authors calculating ice thickness? This is a major finding that makes its way into the abstract, but I wasn't able to find anywhere in the manuscript where this is explained. The PCM calculates a mass loading, but to convert to a thickness requires the authors to assume a packing density. Given that both ash and snow/ice can have wildly different packing densities depending on all sorts of things (distance from the vent, presence of rain at deposition, crystal/clast size, etc.), this decision of density is somewhat critical to the author's results and should be backed up accordingly.

We added a more detailed explanation of how the thickness of ice is calculated in Section 3.1.1 lines 439–445: The precipitation of water is added to the total surface water thickness (expressed in kg m^{-2} and, in the case of ice deposition, can be converted to meters by dividing by the density of ice, which is 920 kg m^{-3}) at a precipitation rate (in $\text{kg m}^{-2} \text{ s}^{-1}$) calculated using the settling velocity of particles assumed to be equal to the terminal fall velocity that we approximate by Stokes law, with a 'slip-flow' correction (Rossow, 1978). In the absence of precipitation, frost (i.e., ice deposition when the atmospheric relative humidity is high) may accumulate and contribute to the total thickness of surface ice.

Since the model outputs the ice precipitation in $\text{kg m}^{-2} \text{ s}^{-1}$ and surface ice in kg m^{-2} , we decided to update the units in figures 2–6, supplementary figures 1–2, the units for surface ice thickness

and ice precipitation throughout the results section, and the units in the extended data table to clear up any potential confusion. We also include ice precipitation rates in both $\text{kg m}^{-2} \text{s}^{-1}$ and m s^{-1} , surface ice loading in kg m^{-2} , and ice thickness in meters throughout the results section so that it is easier for the reader to compare values in the main text to the figures and extended data tables. while also gaining better intuition about how much precipitation/surface ice deposition is occurring.

In lines 150–152 we state: The highest rates of precipitation occur near the eruptive vent, averaging $0.005 \text{ kg m}^{-2} \text{ s}^{-1}$ ($5.4 \times 10^{-6} \text{ m s}^{-1}$ or about 0.5 m sol^{-1} assuming an ice density of 920 kg m^{-3}) under baseline conditions (Fig. 2a; Extended Data Table 1).

In lines 156–159 we state: The modeled ice distribution broadly agrees with areas of excess H, with maximum deposition of $1,340\text{--}1,430 \text{ kg m}^{-2}$ ($\sim 1.5\text{--}1.6 \text{ m}$; Extended Data Table 1) predicted around Apollinaris Mons and Syrtis Major under baseline conditions.

In lines 226–228: However, lower plume heights cause ice to fall to the surface more readily, resulting in increased precipitation rates and up to $\sim 4,200 \text{ kg m}^{-2}$ or $\sim 4.6 \text{ m}$ of surface ice in the case of a 3-sol eruption at a water mass eruption rate of 10^9 kg s^{-1} and a plume height of 35 km (Extended Data Table 1).

In lines 251–258: These smaller grain sizes result in slightly less ice precipitation on average and slightly thinner ice deposits around the volcano by the end of the eruption, whereas low and moderate amounts of CCN (i.e., 104 and 105 kg^{-1}) cause ice to be distributed among fewer nuclei, resulting in larger grain sizes and slightly higher average precipitation rates and ice thicknesses. For example, around Apollinaris Mons, precipitation rates are $4 \times 10^{-3} \text{ kg m}^{-2} \text{ s}^{-1}$ ($4.4 \times 10^{-6} \text{ m s}^{-1}$) for $\text{CCN}=106 \text{ kg}^{-1}$ versus $5 \times 10^{-3} \text{ kg m}^{-2} \text{ s}^{-1}$ ($5.4 \times 10^{-6} \text{ m s}^{-1}$) for $\text{CCN}=104$ and 105 kg^{-1} , with corresponding maximum accumulations of $\sim 1,200 \text{ kg m}^{-2}$ ($\sim 1.3 \text{ m}$) and $\sim 1,340 \text{ kg m}^{-2}$ ($\sim 1.5 \text{ m}$), respectively (Extended Data Table 1).

And finally in lines 277–280: This climate change leads to enhanced protection of surface ice against sublimation even after the volcano has stopped erupting, lowering its sublimation rate to $1.3 \text{ kg m}^{-2} \text{ yr}^{-1}$ ($1.4 \times 10^{-3} \text{ m yr}^{-1}$) compared to $14 \text{ kg m}^{-2} \text{ yr}^{-1}$ ($15 \times 10^{-3} \text{ m yr}^{-1}$) in the same eruption but without H_2SO_4 present (Supplementary Fig. 2).

5. Some volcanological considerations regarding plume heights and mass eruption rates. I've bolded these in the line items (155-163 and 490-501). But in general, I have some concerns with how the authors are relating mass eruption rates and plume heights (especially those that aren't super likely) and how the authors are ignoring the effects of column collapse, especially in very large eruptions as a significant portion of ash would likely not be airborne in that case. *When it comes to column collapse, it is uncertain if it would also occur under early Mars conditions as it does on Earth, given uncertainties in the early atmospheric profile, Mars' lower temperatures, and lower gravity.*

We have added some more detailed discussion regarding the issue of column collapse to Section 3.3.1 lines 600-627: However, it is uncertain if column collapse would also occur under similar conditions on early Mars, given the reduced gravity and uncertainties in the atmospheric density and temperature profile. The atmospheric temperature of early Mars remains debated and may have been colder than Earth, even at similar atmospheric pressures, due to Mars receiving less solar energy because of its greater orbital distance and the faintness of the young Sun (Wordsworth, 2016). Magma erupting into colder air experiences a greater temperature contrast compared to magma erupting into warmer air, which, all else being equal, can drive convecting Plinian columns to higher altitudes. Moreover, the planet's lower gravity leads to higher columns and makes column collapse less likely for the same eruption energy and particle load. Even if column collapse and ignimbrite formation does occur, water present in the column would still get released into the atmosphere and may contribute to surface ice deposits. Moreover, collapsing columns often generate co-ignimbrite plumes, which form above pyroclastic density currents after they sediment coarse clasts, leaving fine particles, volcanic gases, and heated entrained ambient air to lift-off and ascend buoyantly. These plumes can rise nearly as high as the main eruption column, as documented in notable eruptions, such as Toba (75 ka), Tambora (1815), Mount St. Helens (1980), Pinatubo (1991), and the Campanian Ignimbrite eruption (~40 ka) (Woods and Wohletz, 1991; Self and Blake, 2008; Calder et al., 1996; Marti et al., 2016). In some cases, co-ignimbrite plumes may persist for longer durations and introduce more tephra and volatiles into the atmosphere than in the initial Plinian phase, with a mass eruption rate comparable to that of the Plinian stage (Marti et al., 2016). Our modeling approach accounts for this possibility by assuming that most of the material from a co-ignimbrite cloud (if it occurs) is injected into the atmosphere with a mass eruption rate and column height similar to a Plinian eruption. However, if a collapsed column deposits substantial ice near the vent, our model may underestimate proximal ice deposition.

Given the uncertainty surrounding the atmospheric profile of early Mars, the plume height reached for a given mass eruption rate is also not well constrained. Although this issue and column collapse dynamics warrant further study, here we opt to decouple mass eruption rate and plume height to explore their effects on ice distribution independently. We conduct several additional runs to test plume heights of 35, 45, and 65 km for each mass eruption rate.

Line Items:

1. Line 15: Consider adding “which” after “regions” and replacing “containing” with “contain” to make it clearer that the authors are referring to existing measurements. *We have made the suggested change in lines 15–16 now state: This process may have supplied ice to equatorial regions, which contain high excess hydrogen and potential buried ice deposits.*
2. General introduction thoughts: There's a lack of motivations in the intro besides asking what the origin of ice is. The authors could potentially lean into In Situ Resource Utilization (ISRU) as a means of appealing outside the planetary community (given the choice of journal). *Thank you for this suggestion. We think it will be a great addition to the paper. We have added a sentence in the intro in lines 46–48: These findings imply*

there could be an “oasis” of bulk ice in the equatorial regions — an unexpected result that could have significant implications for future human exploration.

We also add a brief discussion about ISRU and broader impacts of this work in the conclusion paragraph. The final paragraph of the conclusion (lines 335–338) now reads: These insights about near-surface ice associated with volcanism and volcanic regions have significant implications for in-situ resource utilization during future human exploration, potential sites for sample return missions⁸⁴, and astrobiology, as ice deposits and volcanic areas are considered promising habitats for microbial life on Mars (Garcia-Lopez et al., 2022; Vishnivetskaya et al., 2022; Khuller et al., 2024; Hadland et al., 2024).

3. Lines 26-38: It may serve to do a more extensive literature review with regards to the anomalous equatorial hydration signals, as there’s a lot of hypotheses that aren’t mentioned here that have significant scientific merit (e.g., hydrous alteration in an aqueous environment). Check out Karunatillake et al. (2014) or Hood et al. (2019) as some starting points. *We have included hypotheses from Karunatillake et al. (2014) or Hood et al. (2019) as suggested. Lines 32–39 now reads: Several possibilities exist for the presence of excess hydrogen: (1) adsorbed water onto regolith particles (Feldman et al., 2004, Malakhov et al., 2020), (2) water incorporated into the mineral’s crystal structure (i.e., hydrated minerals; Malakhov et al., 2020), (3) OH and H₂O located in the structure of salt hydrates (Basilevsky et al., 2003), (4) small amounts of water ice in the pores between regolith particles (Malakhov et al., 2020), (5) hydrous alteration in an aqueous environment (Hood et al., 2019), (6) sulfate hydration in the shallow subsurface (Karunatillake et al., 2014), (7) OH that is part of the structure of clays and trapped water between clay layers (Feldman et al., 2004), and/or (8) water interacting with cations located in the pores of zeolite mineral structure; Feldman et al., 2004).*
4. Lines 28-30: How deep are these instruments sensitive to? It would be good to give that information somewhere. I know GRS is a couple decimeters, but I’m not sure what FRIEND’s sensitivity is. *We have now included this information in the text. Lines 29–32 now reads: However, epithermal neutron data from the Mars Odyssey Neutron Spectrometer (MONS) (Evans et al., 2006; Feldman et al., 2004; Wilson et al., 2018), the Mars Odyssey High Energy Neutron Detector (HEND) (Mitrofanov et al., 2003), and the ExoMars Trace Gas Orbiter’s Fine Resolution Epithermal Neutron Detector (FRIEND) (Malakhov et al., 2020) reveal excess hydrogen in the upper meter of the surface in equatorial regions (between $\pm 30^\circ$).*
5. Line 36: Be careful with the term “permafrost” here. I believe it has a very specific definition on Earth that may not match what the authors are describing here. Calling it “bulk ice” or “ice-ash aggregate” may be better. *We agree with this suggestion and have changed our language to bulk ice in lines 46.*
6. Lines 53-55: Hard to imagine an explosive eruption without ash. Consider removing “if accompanied by ash” and adding “also” after “may”. *We have made the suggested change. Lines 64–66 now read: The result may also be a deposit of an ash-ice mixture, or a layer of ice covered in ash, which may affect how long it remains.*

7. Line 58-59: More volatiles? Different volatiles? *We clarified lines 68–71 to now read: . On Mars, the morphology of volcanic landforms suggest that explosive eruptions were more common in the planet’s early geologic history before transitioning to more effusive eruptions, likely due to a wetter early mantle and/or interactions with external ice or liquid water present on the early Mars surface or near-surface (Robbins et al., 2011; Brož et al., 2021; Kremer et al., 2019; Michalski and Bleacher, 2013).*
8. Lines 59-70: All of the processes listed here don’t usually get preserved in the terrestrial geologic record except in rare circumstances (e.g. raindrop imprints or fulgurites). The authors may need to more explicitly explain why they cite these examples. *Thank you for catching this error, we have mistakenly made reference to the geologic record and we now make it clear that we focus on how these deposits may have influenced the Martian surface. We also point to a few terrestrial examples where this phenomenon has been observed.*
Lines 80–86 now reads: Since most ash particles are effective ice-forming nuclei (Durant et al., 2008), volcanic clouds and subsequent precipitation could have played a key role in shaping Mars’ surface by delivering ash-ice mixtures or ice layers covered in ash. This ash can insulate and preserve underlying ice (discussed further in Section 2.3), a phenomenon observed at Mount St. Helens, Mount Ruapehu in New Zealand, and Villarrica Volcano in Chile (Ayrís and Delmelle, 2012). Additionally, the accumulation of ash over snow or ice-covered soils can promote ground ice formation, as seen after the 2000 Hekla eruption in Iceland (Ayrís and Delmelle, 2012).
9. Lines 97-98: Here and elsewhere, it would be great to refer readers to a context map to get a better sense of where all the features the authors have been mentioning are and which parts are hydrated. *Good suggestion, we have created a global map with the MONS water equivalent hydrogen map overlaid onto a MOLA shaded relief map (Figure 1). We have included two insets that zoom in on Meridiani Planum/Syrtis Major and the MFF/Apollinaris Mons.*
10. Line 102: Consider rephrasing to: “While other workers(46) used volcanic ash deposition as a proxy for volcanic water enhancement in regolith, we explicitly model the release of water into the atmosphere as the result of an explosive eruption while tracking ice deposition”. *We have rephrased this sentence as suggested in lines 122–124.*
11. Line 125: Where is this thickness coming from? How is it being calculated? I can’t find anywhere in the methods that explains this. See main comment above. *We address this point under your main comment (#4) above.*
12. Line 124-125: Was there any sort of statistical link found between the modeled results and the excess hydrogen data? Something beyond qualitative observations. *We’ve remade the maps to highlight the correlation in a more visually apparent way. We will consider statistical correlations in future work.*
13. Line 128: Climatologically analogous I assume. *Yes this is correct. Lines 152–154 now reads: For context, rates of snowfall across the continent of Antarctica, a region that is climatologically analogous to cold conditions likely present on early Mars, can reach ~0.02–0.5 m yr⁻¹(Cuffey and Patterson, 2010).*

14. Lines 130-137 : Citation for these? Or figure callouts? *We restructured some of the text and now discussion of winds are included in the plume height results (lines 216–232) and figure 5.*
15. Figure 1 (and others): The colorscale on most of these figures is confusing to me. Is it logarithmic? Sometimes? It's misleading to make the color breaks at these inconsistent intervals as that may highlight certain features while downplaying others. I would suggest just making the colorscale fully logarithmic. *As stated above in the broad feedback, we have updated figures 2–7 with logarithmic scales.*
16. Also: Given the timescale listed for panel d, are we to assume panels b and c are immediately after eruptions? Please list that somewhere in the figure or in the figure caption. *That is correct, we have updated the figure caption (now figure 2) to read: Combined ice precipitation and surface ice distributions from Syrtis Major and Apollinaris Mons broadly overlap with equatorial regions containing high excess hydrogen (i.e., >10 wt% WEH; red lines). Eruptions from Syrtis Major (67.17°E, 9°N) and Apollinaris Mons (174.4°E, -8.5°S) are run according to baseline parameters listed in Table 2. (a–b) Ice precipitation and surface ice thickness represent the resulting distributions at the end of the eruption. (c) Surface ice thickness represents the resulting distribution one year post-eruption with present-day topography and (d) pre-Tharsis bulge topography. Modeled distributions are overlaid onto a cylindrical projection of Mars and black lines represent MOLA (a–c) topographic contours and pre-Tharsis bulge (d) topographic contours (in kilometers).*
17. Line 143 and elsewhere: Why are these figures in the supplemental if the concept is important enough to be a main section in the results section? Maybe some of these sections belong in a special model sensitivity section? *This is largely a result of having a limited number of allowable display items when we first submitted to Nature Astronomy. Now that we're aiming to publish in Communications we have more leeway with the number of figures and have decided to put the mass eruption rate (figure 3), duration (figure 4), and plume height (figure 5) figures in the main text.*
18. Lines 155-163: Can the authors also comment on the validity of really high MERs but low plume heights? Those don't tend to scale like that (see Mastin et al., 2009 and 2014). *There is great uncertainty in the atmospheric profile of early Mars and therefore the dynamics of Martian column collapse versus convecting Plinian eruptions. Therefore, to fully explore the effect of mass eruption rates on Mars, we opt to do several additional runs in which the mass eruption rate and plume height are decoupled. Lines 218–221 now read: Given the behavior of plumes under the Martian atmosphere may behave differently compared to terrestrial plumes, we decouple the mass eruption rate and plume height and test the sensitivity of ice distribution to each parameter separately (discussed further in Section 3.3.1).*
19. Line 163: Why does the model crash if there's water at high altitude? *We have clarified lines 229–232 to now read: The limits of the model are reached in particularly large eruptions with high plume heights (i.e., a water mass eruption rate of 10^9 kg s⁻¹ and a plume height of 65 km), as the rapid decrease in atmospheric temperatures caused by the high influx of water at such altitudes—and therefore low atmospheric pressures—exceeds the model's physical capabilities.*

20. Lines 172-174: That's neat! *Yes, very interesting!* :D
21. Line 199: There's an argument to be made here of very powerful yet sulfur-weak eruptions, such as the 2022 Hunga Tonga eruption which was largely phreatomagmatic and low ash and sulfur. Given that early Mars had a lot of water (Carr and Head III, 2003; Carr and Head, 2010; Scheller et al., 2021; Boynton et al., 2002; Squyres and Carr, 1986) for erupting magma to interact with, phreato eruptions likely occurred. Not saying the authors are wrong for including the effects of sulfur, just food for thought when making final conclusions of how much of a role sulfur played in ice deposition from eruptions.
- Because of how the paper is structured (Methods coming after the results), we realized our justifications for including sulfuric acid are not clear by the time the reader reaches the results section. *We have moved our justification for the inclusion of sulfuric acid to lines 128–134. It reads: We also simulate the release of sulfuric acid (H₂SO₄) from volcanic eruptions to examine its impact on surface ice deposition. This approach is particularly relevant given that H₂SO₄ is a byproduct of some terrestrial eruptions and can have significant climatic effects (Self et al., 1996; Robock, 2000; LeGrande et al., 2016). Furthermore, observations have shown that Meridiani Planum and the MFF are uniquely enriched in sulfur compared to the bulk average of the Martian regolith (Gaillard et al., 2013; Ojha et al., 2019), suggesting explosive eruptions on Mars may have also expelled significant amounts of sulfur, some of which may have reacted with water to form H₂SO₄ aerosols.*
22. Lines 251-255: The amount of ice being deposited is on the scale of a few meters right? Can these RADAR sounders resolve units that thin? *The radar sounders cannot resolve units that thin, but they see a deposit that is consistent with a high proportion of water ice. To avoid confusion, we have removed this part of the sentence. In lines 299–301 the sentence now simply reads: The resulting stratigraphy from repeated explosive eruptions throughout the lifetime of Apollinaris Mons and Syrtis Major may therefore contain ice-rich layers capped by ice-poor material.*
23. Figure 4: Consider adding some statistics to this figure (e.g. average, min/max, etc.) or to the text that references the figure. *We have added a planetary averaged surface temperature to the caption of the figure (now figure 8). It now reads: The dispersal of H₂SO₄ in the year following an eruption from Apollinaris Mons on average leads to surface cooling across the planet compared to simulations with no H₂SO₄ present. The planetary average surface temperature decreases from 228 K in the control simulation (no eruption) to 217 K in the year following an eruption involving H₂ SO₄ . In contrast, eruptions without H₂ SO₄ result in a planetary average surface temperature of 230 K, only slightly warmer than the control simulation. The eruption simulations are run under baseline conditions as listed in Table 2, except with a water mass eruption rate of 10⁷ kg s⁻¹. In the case with H₂SO₄ present, the H₂SO₄ mass eruption rate is 10⁵ kg s⁻¹. Temperatures are overlaid onto a cylindrical projection of Mars and black lines represent topographic contours (in kilometers).*
24. Also, a figure showing sublimation rate across the planet through time would be neat and help bring home the point about these rates being a dynamic thing. *We have added a figure as described to Supplementary Fig. 2.*

25. Line 287: Consider replacing “code” with “calculations” and deleting “the” before “radiative”. *We have made the suggested changes in line 358.*
26. Line 308-310: Did the authors perform any sensitivity tests with particles that are not spherical? If not, can they comment on what effect nonspherical particles would have on their results given that ash is usually in quite unusual shapes? *We have not, as the model is not currently equipped with this type of particle scheme. In Section 3.1 lines 415–419 we included this discussion: In the PCM, erupted water is represented as spherical particles that are advected by winds and settle to the surface based on their calculated settling velocities. In reality, non-spherical particles may fall more slowly than spherical ones of the same radius, potentially resulting in longer atmospheric residence times than estimated here (Saxby’s et al., 2018).*
27. Line 315: An explanation of what cloud condensation nuclei is/controls would be helpful here. *We have added an explanation of what cloud condensation nuclei is in lines 369–373: We do not explicitly model convective plumes or volcanic ash within the PCM. Instead, the effects of volcanic ash are represented indirectly through cloud condensation nuclei (CCN), (i.e., water-attracting particles) and ice albedo parameters, as atmospheric impurities like ash can promote cloud formation and alter the albedo of surface ice (discussed further in Section 3.3.2).*
28. Line 319-321: Why? Is this a short process? A citation to a study of wet ash aggregation may be necessary. Van Eaton et al. (2015) and Textor et al. (2006ab) may be good places to start. *We edited this sentence to better clarify the message we were trying to communicate. In lines 399–402 we write: We expect that modeling the details of aggregation would change the extent and thickness of the deposit by producing ice distributions that are closer to the source volcano due to the greater settling rate of ash-ice, ash-sulfur, or ice-sulfur aggregates relative to ice alone (e.g., Van Eaton et al., 2015; Textor et al., 2006b).*
29. Line 338: CCN should be defined earlier. *We have defined CCN earlier in the previous section as suggested and use section 3.1.1 to elucidate how we parametrize the cloud particle radii.*
30. Line 338: What does “activated” mean? *We clarified the meaning of “activated” in lines 421–422: When water vapor in the air condenses onto these particles, they grow into cloud droplets, effectively “activating” the CCNs)*
31. Line 339: It’s never explained in section 4.5.2 how the authors calculate particle radii from CCN. *We now explain in Section 3.1.1 lines 419–433 how we calculate particle radii: Local H₂O cloud particle radii are based on the amount of condensed material and are determined by the amount of activated cloud condensation nuclei per unit mass of air, N_{mix} (i.e., when water vapor in the air condenses onto these particles, they grow into cloud droplets, effectively “activating” the CCNs). N_{mix} is taken to be constant everywhere in the atmosphere (but varied in sensitivity tests; discussed further in Section 3.3.2) and is equal to 10⁵ kg⁻¹ (particles per kg of air) for both water ice and vapor clouds in the baseline simulation. Following the procedure outlined in previous PCM studies (Hamid et al., 2024; Madeleine et al., 2012; Wordsworth et al., 2013; Forget et al., 2013; and Turbet et al., 2020), the effective radius of the cloud particles is given by: $r = ((3q_c)/(4\pi\rho_c N_{mix}))^{1/3}$*

Where q_c is the mass mixing ratio of cloud particles (in kg kg^{-1} of air), and ρ_c is the density of the cloud particles (1000 kg m^{-3} for liquid water and 920 kg m^{-3} for water ice). The effective radius of the cloud particle is then used to compute their settling velocity and radiative properties calculated by Mie scattering, which occurs when the particle size is comparable to the wavelength of light.

32. Lines 350-352: It doesn't necessarily have to be water from the plume. It could also be environmental. We agree, lines 449–461 now reads: During some terrestrial explosive volcanic eruptions, sulfur dioxide (SO_2) is injected into the stratosphere, where it reacts with water from the eruptive plume or from external water present in the environment to form sulfuric acid (H_2SO_4) aerosols (Self et al., 1996; LeGrande et al., 2016; Zhu et al., 2020; Robock, 2000).

33. Line 362: What is a Mie code? What does that mean? We have clarified the meaning of Mie theory and scattering in Section 3.1 lines 430–437 how the radiative properties of cloud particles are calculated: The effective radii of cloud particles are then used to compute their settling velocities and radiative properties calculated by Mie scattering, which occurs when the particle size is comparable to the wavelength of light. This process involves generating lookup tables of the extinction coefficient (the strength of absorption or reflection at a given wavelength), single-scattering albedo (the ratio of scattered to absorbed light), and the Henyey-Greenstein asymmetry factor (the degree of forward or backward scattering). These tables are then used by the PCM to compute cloud scattering parameters (Bohren and Huffman, 1983).

Finally, in Section 3.1.2 lines 456–460 we explain how Mie scattering is used to calculate the radiative properties of sulfuric acid particles: To investigate the effect of volcanic H_2SO_4 aerosols on surface ice accumulation, we follow the approach of Kerber et al. (2015), implementing a simplified scheme to assess the sensitivity of our results to H_2SO_4 -induced cooling without explicitly modeling the formation process. Using Mie theory (as described in Section 3.1 and 3.1.1), we generate lookup tables for the PCM to calculate cloud scattering parameters of H_2SO_4 in each grid box (Bohren and Huffman, 1983).

34. Line 366: Physically inert too? No, it's physically active and when released can be advected by the wind (e.g., Fig. 3) and is susceptible to gravitational settling. We have clarified this point in the text as well in lines 460–463: To simulate emissions of H_2SO_4 that might be generated during an explosive event, we treat its mass eruption rate as a free parameter and inject it into the model as physically and radiatively active, but chemically inert $0.3\text{-}\mu\text{m}$ spherical tracers (particle size is based on typical terrestrial volcanic aerosol radii; Kerber et al., 2015).

35. Lines 376-380: Given the likely overlap of some of these volcano's ages and Tharsis's existence, it would be nice to see a sensitivity test of removing the Tharsis bulge (an easy knob to turn in the PCM). Thanks for his suggestion. We actually find that including a test pre-Tharsis bulge reveals that the lack of ice accumulation around Tharsis enables greater ice deposition in equatorial regions (Figure 2d).

36. Lines 428-434: Yes, but there's other things to consider here too such as magma ascent rate. There could also just simply be less magma to erupt due to a bunch of factors (local lithology, plate tectonic setting etc.). *Yes, that is true.*
37. Line 435-438: It's important to note here that both these citations give the "Mars has more explosivity" argument due to a current martian atmosphere, not an ancient one like is being simulated here. Consider removing this sentence or specify that it's for current martian pressure. *We address this comment above under the second broad comment.*
38. Lines 435-448: Also consider including the effects of available surface water or ice enhancing the explosivity of a basaltic eruption (Brož et al., 2021; Brož and Hauber, 2013; Sears et al., 2022; Wilson and Head III, 2004; Wilson and MouginisMark, 2003; Paladino et al. 2024). *Thank you for this suggestion, we have added reference to these effects in lines 545–548, which is also mentioned in a previous comment: The presence of abundant surface water on early Mars could further enhance explosivity when magma interacts with external water. As a result, explosive basaltic eruptions on early Mars may have resembled terrestrial Plinian eruptions, reaching heights of tens of kilometers (Woods, 1995; Hort and Weitz, 2001).*
39. Line 450: Think the authors lost a parenthesis here. *Good catch, we have added that lost parenthesis.*
40. Line 461: Does this assumption hold up on Mars (those citations are for terrestrial eruptions)? Martian atmospheric chemistry is awfully different than terrestrial chemistry and the available constituent chemical ratios are likely different. *Thank you for pointing this out — Yes these assumptions hold up if we assume that the early Martian atmosphere was oxidizing. A recently published study by some members in our team found that SO₂ readily converts into H₂SO₄ under an oxidizing early Martian atmosphere (Braude et al., 2025). However, this study also found that in a transient reducing atmosphere S₈ may have formed more preferentially over H₂SO₄, which may have resulted in slight, but mostly negligible warming if the S₈ is very large (i.e., radius of 10 μm), but otherwise results in cooling for smaller particles (i.e., radius of 0.1 and 1 μm).*

In lines 564–570 we state: The atmospheric chemistry of Mars may influence the efficiency of this conversion. Previous work indicates that if early Mars had an oxidizing atmosphere (e.g., predominantly CO₂), H₂SO₄ readily forms from volcanic SO₂ outgassing, while more reducing conditions may have favored the formation of S₈ (Braude et al., 2025). In our model we assume an oxidizing atmosphere and the conversion of SO₂ to H₂SO₄, consistent with the transition from a reducing to an oxidizing atmosphere early in the Noachian era (Liu et al., 2024).

41. Line 482: I think this citation is wrong (should be Glaze & Baloga, 2002). *Good catch, we have corrected this error in line 591.*
42. Lines 490-501: Ok, but the authors are simulating these eruptions in a 1 bar atmosphere which is fairly well established to be a reasonable estimate of martian surface pressure in the Noachian and identical in pressure to modern Earth so column collapse would most likely be occurring at these massive eruption rates. Collapse would also likely be occurring at mass eruption rates orders of magnitude lower than 10¹⁰ or 10¹¹ kg/s. Given this and the fact that the PDCs tend to deposit significant amounts of material

close to the vent (even if they form phoenix clouds), I'm not sure the authors can fully ignore the effects of column collapse. I would suggest at least acknowledging this and providing some back-of-the-envelope calculations of how this would affect the authors' results or (if time and space allow) provide another scenario in which some degree of mass (20%? 30%?) is assumed to be deposited locally around the volcano and running sets of simulations at these reduced mass loads. *We address this point in more detail under the broad comment (#5).*

43. Line 511: Might be worthwhile to cite some ash hydration studies here (e.g. Aomine and Wada, 1962; Seligman et al., 2016; Hudak et al., 2021). *We agree and have added the suggested references as well as Scheller et al., 2021 in line 637*
44. Lines 519-522: Similar to my comments on lines 435-438, the cited source here is for a study on a current martian atmosphere, not an ancient one. The authors might want to revisit their rationale in this section. In an ancient atmosphere, is heterogeneous nucleation still the dominant mechanism? *We address this comment in detail under the second broad comment.*
45. Line 529: Consider putting kg^{-1} after these numbers for consistency purposes. *We have made this change as suggested in line 660.*

Reviewer #3 (Remarks to the Author):

2nd Review of, "Precipitation Induced by Explosive Volcanism on Mars and its Implications for Ice in Unexpected Equatorial Regions"

Broad Thoughts:

This manuscript explores the effects of explosive volcanism as a significant source for the anomalously high amounts of water found in equatorial regions of Mars as measured by remote sensing satellites. The authors use a GCM to simulate ash and water dispersal and explore myriad parameters that could affect the resulting ash/water distribution patterns and longevity, including obliquity, eruption rate, and sulfur content (amongst others). The authors find that significantly large explosive eruptions can produce meters of water ice in the areas that are observed to be enriched in hydrogen.

After the first round of review, the manuscript is looking in great shape. The authors addressed most comments made by myself and other reviewers in a well thought out and convincing manner, adding descriptions or new figures in the text where necessary. Overall, the study is well written, scientifically significant, convincing, and will make an excellent addition to the literature. A few more comments and line items to check out, but I don't need to see the manuscript again.

Great job, Saira! This looks like a ton of work and you've presented it well. Excited to see what you do next!

-Tyler Paladino

Comments:

1. Thank you for adding some information about the assumed density used for thickness calculations. I would still appreciate some commentary on why this value for density was chosen given it can change a lot with amount of ice or distance from vent. For a bunch of the figures that have units of kg/m^2 , the label should read "surface ice mass loading" or something similar rather than "surface ice thickness".

Thank you for this comment. We have updated the labels to read surface ice mass loading and have changed references to surface ice "thickness" to surface ice "loading" throughout the text. We use an ice density of 920 kg m^{-3} as the default in our model, which corresponds to hexagonal ice, which is the most common crystalline form on Earth. Lines 414–417 where we first mention the density of ice now reads: Where q_c is the mass mixing ratio of cloud particles (in kg kg^{-1} of air), and ρ_c is the density of the cloud particles (1000 kg m^{-3} for liquid water and 920 kg m^{-3} for water ice, corresponding to hexagonal ice—the most common crystalline form on Earth, and expected under early Mars conditions).

Line items:

1. Figure 1: Love the context map. Could boxes be added to the panel a that correspond to the inset panels b and c to give an idea of where we're zoomed into? *We have made the requested change.*
2. Figure 3 (and others): Can the authors add a symbol to show where the eruptive vent(s) is on the map? Also, the numbers on the map are labeling the topo contours, right? That could be explicitly written out somewhere in the caption. Otherwise, I could see that causing some confusion. *We have clarified the labeling of the topographic contours in the captions as suggested. However, we chose not to add a symbol for the volcano locations in the figures, as it would obscure portions of the modeled distribution. We instead point the reader to the figure 1 context map and provide the coordinates of the volcano in each of the figure captions.*
3. Figure 5: The quiver plots are a bit incomprehensible right now. Consider removing maybe a quarter to a third of the quivers to make it a bit more obvious which way the wind is blowing. It might also be good to experiment with making all the quivers for each plot be on the same scale. *Thank you for this suggestion. We have improved the figure's readability by increasing the arrow sizes and ensuring that the wind arrows in each plot are drawn to the same scale.*
4. Lines 249-258: I was slightly confused when reading the discussion about grainsizes. The grain size the authors talk about is the ice/ash aggregate grainsize correct? Is this explicitly modeled or inferred? I didn't think the PCM could deal with actively changing grainsizes (but I might be wrong).

The PCM does not explicitly model ice-ash aggregates. Rather, the particles (the PCM is agnostic to the composition of these particles and just recognizes the presence of a particle that clouds can condense onto) are introduced as a concentration of "activated cloud condensation nuclei", or CCN. For each grid cube, the amount of water in the cube is divided among the CCN, establishing a water droplet size. The size or density of the nucleating grain is not captured in the model.

We have added clarification in lines 221–224 explaining how cloud radii are calculated, referencing the Methods: "Local H₂O cloud particle radii are based on the amount of condensed material, which is determined by the number of CCN per unit mass of air. Terminal fall velocities of these particles are then calculated from their radii and densities (see Methods: Water Precipitation and Deposition)."

Additionally, we have replaced "grain size" with "particle size" throughout this paragraph for improved clarity.

5. Lines 277-287: What timescale did the authors find for these protection effects to go away?

We did not model the longer-duration post-eruption simulations to determine exactly when the protective effects dissipate, due to the high computational cost and large data volumes required at this resolution. Since there is no sulfuric acid currently in the atmosphere, these aerosols will eventually settle out, and we instead opt to discuss the general effects of their eventual removal.

6. Lines 314-334: I'm still feeling a bit wary on this idea. I think it's ok to say that there were explosive eruptions as atmospheric pressure decreased, but the way the eruptions would behave would be different and wouldn't necessarily play by the rules or assumptions of normal plumes like are used in this study. I think just mentioning that somewhere in this section would be sufficient: That volcanoes in this low-pressure time periods may indeed have contributed to ice deposits, but that the way eruptions are modeled in this study is fundamentally different than how they would behave in a low-pressure atmosphere.

Thank you for this suggestion. We do not explicitly model plume dynamics in this study; instead, we use release heights loosely based on buoyant rise in a Mars-like atmosphere and then simulate atmospheric dispersal. Whether the eruption behaved as a plume or a jet is therefore not critical—only the release height matters. Since atmospheric pressures during the periods of ice deposition are poorly constrained, we explore a range of release heights expected to encompass plausible plume heights instead of modeling them directly.

7. Lines 534-550: Thank you for addressing this. Might be worthwhile to tack on a bit to the end to acknowledge that plumes in a thinner atmosphere would behave pretty differently (see Wilson and Head, 2007 for their “inertial plumes” idea)

Thank you for this suggestion. Lines 540–544 now read: As Mars transitioned into the Hesperian and atmospheric pressure declined (Scherf and Lammer, 2021), the thinning atmosphere—combined with the planet's low gravity—may have further amplified the explosivity of volcanic eruptions, potentially driving plumes into an inertial regime, behaving more like jets in which pyroclasts travel ballistically, governed primarily by inertia and gravity (Wilson and Head, 2007).

8. Lines 604-607: This needs a citation. *We have added the citation to Glaze and Baloga, 1996 in line 602.*

9. Lines 600-627: Thanks for adding this section! Adds some much needed discussion/caveats on the topic. Though I might push back a little on the “uncertainty of column collapse occurring in early Martian conditions” claim. Column collapse would definitely be happening at the modeled atmospheric pressures and gravity (there's a handful of geologic examples), but the degree to which it occurs within a single eruption is uncertain (i.e. how much material is collapsing). It's hard to have a plume that doesn't have some amount of material that's collapsing, especially at really high MERs.

We agree that column collapse could occur on Mars under a denser atmosphere. Our point is that it remains uncertain whether it would occur under the same conditions as on Earth (i.e., total mass eruption rates $>10^9$ kg/s), given the reduced gravity and the uncertainties in Mars' atmospheric density and temperature profile.